# SUCCESSOR FEATURE REPRESENTATIONS

**Chris Reinke & Xavier Alameda-Pineda**
RobotLearn
INRIA Grenoble, LJK, UGA
{chris.reinke,xavier.alameda-pineda}@inria.fr

## ABSTRACT

Transfer in Reinforcement Learning aims to improve learning performance on target tasks using knowledge from experienced source tasks. Successor Representations (SR) and their extension Successor Features (SF) are prominent transfer mechanisms in domains where reward functions change between tasks. They reevaluate the expected return of previously learned policies in a new target task to transfer their knowledge. The SF framework extended SR by linearly decomposing rewards into successor features and a reward weight vector allowing their application in high-dimensional tasks. But this came with the cost of having a linear relationship between reward functions and successor features, limiting its application to tasks where such a linear relationship exists. We propose a novel formulation of SR based on learning the cumulative discounted probability of successor features, called Successor Feature Representations (SFR). Crucially, SFR allows to reevaluate the expected return of policies for general reward functions. We introduce different SFR variations, prove its convergence, and provide a guarantee on its transfer performance. Experimental evaluations based on SFR with function approximation demonstrate its advantage over SF not only for general reward functions, but also in the case of linearly decomposable reward functions.

## PREFACE

This version of the paper is presented at the Reincarnating Reinforcement Learning (RRL) Workshop at ICLR 2023. Its final version has been accepted at Transactions for Machine Learning Research and can be accessed via: `https://openreview.net/pdf?id=MTFf1rDDEI`.

The RRL workshop discusses approaches that reuse prior computation in RL. In alignment with this goal, this paper discusses a new variant of transfer learning for RL tasks with changing reward functions. It allows to reuse policies learned for past tasks (reward functions) be reevaluating them for a new tasks and to transfer previously learned behaviors if this is beneficial.

## 1 INTRODUCTION

One of the goals of Artificial Intelligence (AI) is to design agents that have the same abilities as humans, including to quickly adapt to new environments and tasks. Reinforcement Learning (RL), the branch of AI that learns new behaviors based on the maximization of reward signals from the environment, has already successfully addressed many complex problems such as playing computer games, chess, and even Go with superhuman performance (Mnih et al., 2015; Silver et al., 2018). These impressive results are possible thanks to a vast amount of interactions of the RL agents with their environment/task during learning. Nonetheless, such a strategy is unsuitable for settings where the amount of environment interactions is costly or restricted, for example, in robotics or when an agent has to adapt quickly to a new task or environment. Consider a caregiver robot in a hospital that has to learn a new task, such as a new route to deliver meals. In such a setting, the agent can not collect a vast amount of training samples but has to adapt quickly. Transfer learning aims to provide mechanisms to address this issue (Taylor & Stone, 2009; Lazaric, 2012; Zhu et al., 2020). The rationale is to use knowledge from previously encountered source tasks for a new target task to improve the learning performance on this target task. The previous knowledge can help reduce the amount of interactions required to learn the new optimal behavior. For example, the caregiver robot

could reuse knowledge about the layout of the hospital learned in previous source tasks (e.g. guiding a person) to learn to deliver meals.

The Successor Feature (SF) and General Policy Improvement (GPI) framework (Barreto et al., 2020) is a prominent transfer learning mechanism for tasks where only the reward function differs. SF is an extension of Successor Representations (SR) (Dayan, 1993). SR represents a policy (a learned behavior) by decoupling its dynamics from the expected rewards. The dynamics are described by the cumulative discounted probability of successor states for a start state $s$, i.e. a description of which future states the agent will visit if using the policy starting from $s$. This allows evaluating the policy by any reward function that defines rewards over visited states.

The SF framework extended SR by allowing high-dimensional tasks (state inputs) and by introducing the GPI procedure for the transfer of knowledge between tasks. Its basic premise is that rewards $r = R(\phi)$ are defined based on a low-dimensional feature vector $\phi \in \mathbb{R}^n$ that describes the important features of a high-dimensional state. For our caregiver robot, this could be ID's of beds or rooms that it is visiting, in difference to its high-dimensional visual state input from a camera. The rewards are then computed not based on its visual input but on the ID's of the beds or rooms it visits. The expected cumulative discounted successor features ($\psi$) are learned for each policy that the robot learned in the past. It represents the dynamics in the feature space that the agent experiences for a policy. This corresponds to the rooms or beds the caregiver agent would visit if using the policy. This representation of feature dynamics is independent of the reward function. A behavior learned in a previous task and described by this SF representation can be directly re-evaluated for a different reward function. In a new task, i.e. for a new reward function, the GPI procedure re-evaluates the behaviors learned in previous tasks for it. It then selects at each state the behavior of a previous task if it improves the expected reward. This allows reusing behaviors learned in previous source tasks for a new target task. A similar transfer strategy can also be observed in the behavior of humans (Momennejad et al., 2017; Momennejad, 2020; Tomov et al., 2021).

The SF&GPI framework (Barreto et al., 2017; 2018) makes the assumption that rewards are a linear composition of the features $\phi \in \mathbb{R}^n$ via a reward weight vector $\mathbf{w}_i \in \mathbb{R}^n$ that depends on the task $i$: $r_i = \phi^\top \mathbf{w}_i$. This assumption allows to effectively separate the feature dynamics of a behavior from the rewards and thus to re-evaluate previous behaviors given a new reward function, i.e. a new weight vector $\mathbf{w}_j$. Nonetheless, this assumption also restricts the successful application of SF&GPI only to problems where such a linear decomposition is possible or can be approximated.

We propose a new formulation of the SR framework, called Successor Feature Representations (SFR), to allow the usage of general reward functions over the feature space: $r_i = R_i(\phi)$. SFR represents the cumulative discounted probability over the successor features, referred to as the $\xi$-function. We introduce SFRQ-learning (SFRQL) to learn the $\xi$-function. Our work is related to Janner et al. (2020); Touati & Ollivier (2021), and brings two important additional contributions. First, we provide mathematical proof of the convergence of SFRQL. Second, we demonstrate how SFRQL can be used for meta-RL, using the $\xi$-function to re-evaluate behaviors learned in previous tasks for a new reward function $R_j$. Furthermore, SFRQL can also be used to transfer knowledge to new tasks using GPI.

The contribution of our paper is three-fold:

1. We introduce a new RL algorithm, SFRQL, based on a cumulative discounted probability of successor features.

2. We provide a theoretical proofs of the convergence of SFRQL to the optimal policy and for a guarantee of its transfer learning performance under the GPI procedure.

3. We experimentally compare SFRQL in tasks with linear and general reward functions, and for tasks with discrete and continuous features to standard Q-learning and the classical SF framework, demonstrating the interest and advantage of SFRQL.

## 2 BACKGROUND

### 2.1 REINFORCEMENT LEARNING

RL investigates algorithms to solve multi-step decision problems, aiming to maximize the sum over future rewards (Sutton & Barto, 2018). RL problems are modeled as Markov Decision Processes (MDPs) which are defined as a tuple $M \equiv (\mathcal{S}, \mathcal{A}, p, R, \gamma)$, where $\mathcal{S}$ and $\mathcal{A}$ are the state and action set. An agent transitions from a state $s_t$ to another state $s_{t+1}$ using action $a_t$ at time point $t$ collecting a reward $r_t$: $s_t \xrightarrow{a_t, r_t} s_{t+1}$. This process is stochastic and the transition probability $p(s_{t+1}|s_t, a_t)$ describes which state $s_{t+1}$ is reached. The reward function $R$ defines the scalar reward $r_t = R(s_t, a_t, s_{t+1}) \in \mathbb{R}$ for the transition. The goal in an MDP is to maximize the expected return $G_t = \mathbb{E}\left[\sum_{k=0}^{\infty} \gamma^k R_{t+k}\right]$, where $R_t = R(S_t, A_t, S_{t+1})$. The discount factor $\gamma \in [0, 1)$ weights collected rewards by discounting future rewards stronger. RL provides algorithms to learn a policy $\pi : \mathcal{S} \to \mathcal{A}$ defining which action to take in which state to maximise $G_t$.

Value-based RL methods use the concept of value functions to learn the optimal policy. The state-action value function, called Q-function, is defined as the expected future return taking action $a_t$ in $s_t$ and then following policy $\pi$:

$$Q^\pi(s_t, a_t) = \mathbb{E}_\pi \left\{ r_t + \gamma r_{t+1} + \gamma^2 r_{t+2} + \ldots \right\} = \mathbb{E}_\pi \left\{ r_t + \gamma Q^\pi(S_{t+1}, A_{t+1}) \right\} . \tag{1}$$

The Q-function can be recursively defined following the Bellman equation such that the current Q-value $Q^\pi(s_t, a_t)$ depends on the Q-value of the next state $Q^\pi(s_{t+1}, a_{t+1})$. The optimal policy for an MDP can then be expressed based on the Q-function, by taking at every step the maximum action: $\pi^*(s) \in \arg\max_a Q^*(s, a)$.

The optimal Q-function can be learned using a temporal difference method such as Q-learning (Watkins & Dayan, 1992). Given a transition $(s_t, a_t, r_t, s_{t+1})$, the Q-value is updated according to:

$$Q_{k+1}(s_t, a_t) = Q_k(s_t, a_t) + \alpha_k \left( r_t + \max_{a_{t+1}} Q_k(s_{t+1}, a_{t+1}) - Q_k(s_t, a_t) \right) , \tag{2}$$

where $\alpha_k \in (0, 1]$ is the learning rate at iteration $k$.

### 2.2 SUCCESSOR REPRESENTATIONS

Successor Representations (SR) were introduced as a generalization of the value function (Dayan, 1993). SR represents the cumulative discounted probability of successor states for a policy. Its state-action representation (White, 1996) is defined as:

$$M^\pi(s_t, a_t, s') = \sum_{k=0}^{\infty} \gamma^k p(S_{t+k} = s'|s_t, a_t; \pi) , \tag{3}$$

where $p(S_{t+k} = s'|s_t, a_t; \pi)$ is the probability of being in state $s'$ at time-point $t + k$ if the agent started in state $s_t$ using action $a_t$ and then following policy $\pi$. Given a state-dependent reward function $r_t = R(s_t)$, the Q-value, i.e. the expected return, can be computed by:

$$Q^\pi(s, a) = \sum_{s' \in S} M^\pi(s, a, s') R(s') . \tag{4}$$

This allows to evaluate the expected return of a learned policy for any given reward function. The SR framework has been restricted to low-dimensional and discrete state space because the representation of $M$ (3) and the sum operator in the Q-value definition (4) are difficult to implement for high-dimensional and continuous states.

### 2.3 TRANSFER LEARNING WITH SUCCESSOR FEATURES

The Successor Feature (SF) framework (Gehring, 2015; Barreto et al., 2017; 2018) is an extension of SR to handle high-dimensional, continuous state spaces and to use the method for transfer learning. In the targeted transfer learning setting agents have to solve a set of tasks (MDPs) $\mathcal{M} = \{M_1, M_2, \ldots, M_m\}$, that differ only in their reward function. SF assumes that the reward

function can be decomposed into a linear combination of features $\phi \in \Phi \subset \mathbb{R}^n$ and a reward weight vector $\mathbf{w}_i \in \mathbb{R}^n$ that is defined for a task $M_i$:

$$r_i(s_t, a_t, s_{t+1}) \equiv \phi(s_t, a_t, s_{t+1})^\top \mathbf{w}_i \, . \tag{5}$$

We refer to such reward functions as linear reward functions. Since the various tasks differ only in their reward functions, the features are the same for all tasks in $\mathcal{M}$.

Given the decomposition above, it is also possible to rewrite the Q-function into an expected discounted sum over future features $\psi^{\pi_i}(s, a)$ and the reward weight vector $\mathbf{w}_i$:

$$Q_i^{\pi_i}(s, a) = \mathbb{E}\left\{ r_t + \gamma^1 r_{t+1} + \gamma^2 r_{t+2} + \ldots \right\} = \mathbb{E}\left\{ \phi_t^\top \mathbf{w}_i + \gamma^1 \phi_{t+1}^\top \mathbf{w}_i + \gamma^2 \phi_{t+2}^\top \mathbf{w}_i + \ldots \right\}$$

$$= \mathbb{E}\left\{ \sum_{k=0}^{\infty} \gamma^k \phi_{t+k} \right\}^\top \mathbf{w}_i \equiv \psi^{\pi_i}(s, a)^\top \mathbf{w}_i \, . \tag{6}$$

This decouples the dynamics of the policy $\pi_i$ in the feature space of the MDP from the expected rewards for such features. Thus, it is now possible to evaluate the policy $\pi_i$ in a different task $M_j$ using a simple multiplication of the weight vector $\mathbf{w}_j$ with the $\psi$-function: $Q_j^{\pi_i}(s, a) = \psi^{\pi_i}(s, a)^\top \mathbf{w}_j$. Interestingly, the $\psi$ function also follows the Bellman equation:

$$\psi^\pi(s, a) = \mathbb{E}\left\{ \phi_{t+1} + \gamma \psi^\pi(s_{t+1}, \pi(s_{t+1})) | s_t, a_t \right\} \, . \tag{7}$$

It can therefore be learned with conventional RL methods. Moreover, Lehnert & Littman (2019) showed the equivalence of SF-learning to Q-learning.

Being in a new task $M_j$ the Generalized Policy Improvement (GPI) can be used to select the action over all policies learned so far that behaves best:

$$\pi(s) \in \arg\max_a \max_i Q_j^{\pi_i}(s, a) = \arg\max_a \max_i \psi^{\pi_i}(s, a)^\top \mathbf{w}_j \, . \tag{8}$$

Barreto et al. (2018) proved that under the appropriate conditions for optimal policy approximates, the policy constructed in (8) is close to the optimal one, and their difference is upper-bounded:

$$\|Q^* - Q^\pi\|_\infty \leq \frac{2}{1 - \gamma} \left( \|r - r_i\|_\infty + \min_j \|r_i - r_j\|_\infty + \epsilon \right) \, , \tag{9}$$

where $\|f - g\|_\infty = \max_{s,a} |f(s, a) - g(s, a)|$ and $r_i$ is the reward of the closest task $M_i$ to the current one $M$ that can be linearly decomposed as in (5). Given the current task $M$, and the theoretically closest possible linear reward task $M_i$, we search the linear task $M_j$ in our set of tasks $\mathcal{M}$ (from which we also construct the GPI optimal policy (8)) which is closest to it. The upper bound between $Q^*$ and $Q$ is then defined by 1) the difference between task $M$ and the theoretically closest possible linear task $M_i$: $\|r - r_i\|_\infty$; and by 2) the difference between theoretical task $M_i$ and the closest task $M_j$: $\min_j \|r_i - r_j\|_\infty$. If our new task $M$ is also linear then $r = r_i$ and the first term in (9) would vanish. See Barreto et al. (2018) for more details.

Very importantly, this result shows that the SF framework will only provide a good approximation of the true Q-function if the reward function in a task can be represented using a linear decomposition. If this is not the case, then the error in the approximation increases with the distance between the true reward function $r$ and the best linear approximation of it $r_i$ as stated by $\|r - r_i\|_\infty$. For example, a reward function $R(\phi) = \exp(-(\phi - a)^2)$ using a Gaussian kernel to express a preferred feature around $a$ can not be directly represented.

In summary, SF&GPI extends the SR framework by introducing the concept of features over which the reward function is defined. This makes the transfer of knowledge between tasks with high-dimensional and continuous states possible. However, it introduces the limitation that reward functions are a linear combination of features and a reward weight vector reducing their practical application for arbitrary reward functions.

## 3 METHOD: SUCCESSOR FEATURE REPRESENTATIONS

### 3.1 DEFINITION AND FOUNDATIONS

The goal of this paper is to adapt the SR and SF frameworks to tasks with general reward functions $R : \Phi \mapsto \mathbb{R}$ over state features $\phi \in \Phi$:

$$r(s_t, a_t, s_{t+1}) \equiv R(\phi(s_t, a_t, s_{t+1})) = R(\phi_t) \, , \tag{10}$$

where we define $\phi_t \equiv \phi(s_t, a_t, s_{t+1})$. Under this assumption the Q-function can not be linearly decomposed into a part that describes feature dynamics and one that describes the rewards as in the SF framework (6). To overcome this issue, we propose to use the future cumulative discounted probability of successor features, named $\xi$-function, which is going to be the central mathematical object of the paper, as:

$$\xi^\pi(s, a, \phi) = \sum_{k=0}^{\infty} \gamma^k p(\phi_{t+k} = \phi | s_t = s, a_t = a; \pi) , \qquad (11)$$

where $p(\phi_{t+k} = \phi | s_t = s, a_t = a; \pi)$, or in short $p(\phi_{t+k} = \phi | s_t, a_t; \pi)$, is the probability density function of the features at time $t + k$, following policy $\pi$ and conditioned to $s$ and $a$ being the state and action at time $t$ respectively. Note that $\xi^\pi$ depends not only on the policy $\pi$ but also on the state transition (constant through the paper). With the definition of the $\xi$-function, the Q-function rewrites:

$$Q^\pi(s_t, a_t) = \sum_{k=0}^{\infty} \gamma^k \mathbb{E}_{p(\phi_{t+k}|s_t, a_t; \pi)} \{R(\phi_{t+k})\} = \sum_{k=0}^{\infty} \gamma^k \int_\Phi p(\phi_{t+k} = \phi | s_t, a_t; \pi) R(\phi) \mathrm{d}\phi$$

$$= \int_\Phi R(\phi) \sum_{k=0}^{\infty} \gamma^k p(\phi_{t+k} = \phi | s_t, a_t; \pi) \mathrm{d}\phi = \int_\Phi R(\phi) \xi^\pi(s_t, a_t, \phi) \mathrm{d}\phi . \qquad (12)$$

Both the proposed SFR and the SR frameworks exploit cumulative discounted sums of probability distributions. In the case of SR, the probability distributions are defined over the states (3). In contrast, the $\xi$-function of SFR is defined over features (and not states). There exists also an inherent relationship between SFR and SF. First, SF is a particular case of SFR when constrained to linear reward functions, see Appendix A.4. Second, in the case of discrete features ($\phi \in \mathbb{N}^n$), we can reformulate the SFR problem into a linear reward function that is solved similarly to SF, see Appendix A.5. Therefore, SFR allows to consider these two cases within the same umbrella, and very importantly, to extend the philosophy of SF to continuous features and non-linear reward functions.

## 3.2 SFR Learning Algorithms

In order to learn the $\xi$-function, we introduce SFRQ-learning (SFRQL). Its update operator is an off-policy temporal difference update analogous to Q-learning and SFQL (Barreto et al., 2017). Given a transition $(s_t, a_t, s_{t+1}, \phi_t)$ the SFRQL update operator is defined as:

$$\xi_{k+1}^\pi(s_t, a_t, \phi) = (1 - \alpha_k)\xi_k^\pi(s_t, a_t, \phi) + \alpha_k\Big(p(\phi_t = \phi | s_t, a_t) + \gamma\xi_k^\pi(s_{t+1}, \bar{a}_{t+1}, \phi)\Big), \qquad (13)$$

where $\bar{a}_{t+1} = \arg\max_a \int_\Phi R(\phi)\xi^\pi(s_{t+1}, a, \phi)\mathrm{d}\phi$.

The following theorem provides convergence guarantees for SFRQL under the assumption that either $p(\phi_t = \phi | s_t, a_t; \pi)$ is known, or an unbiased estimate can be constructed, and is one of the main results of this paper:

**Theorem 1.** *(Convergence of SFRQL) For a sequence of state-action-feature $\{s_t, a_t, s_{t+1}, \phi_t\}_{t=0}^{\infty}$ consider the SFRQL update given in (13). If the sequence of state-action-feature triples visits each state, action infinitely often, and if the learning rate $\alpha_k$ is an adapted sequence satisfying the Robbins-Monro conditions:*

$$\sum_{k=1}^{\infty} \alpha_k = \infty, \qquad \sum_{k=1}^{\infty} \alpha_k^2 < \infty \qquad (14)$$

*then the sequence of function classes corresponding to the iterates converges to the optimum, which corresponds to the optimal Q-function to which standard Q-learning updates would converge to:*

$$[\xi_n] \to [\xi^*] \quad with \quad Q^*(s, a) = \int_\Phi R(\phi)\xi^*(s, a, \phi)d\phi. \qquad (15)$$

The proof follows the same flow as for Q-learning and is provided in Appendix A.2. The main difference is that the convergence of SFRQL is proven up to an additive function $\kappa$ with zero reward,

i.e. $\int_\Phi R(\phi)\kappa(s,a,\phi)\mathrm{d}\phi = 0$. More formally, the convergence is proven to a class of functions $[\xi^*]$, rather that to a single function $\xi^*$. This is not problematic, since additive functions with zero reward are meaningless for reinforcement learning.

As stated above, the previous result assumes that either $p(\phi_t = \phi|s_t, a_t; \pi)$ is known, or an unbiased estimate can be constructed. We propose two different ways to approximate $p(\phi_t = \phi|s_t, a_t; \pi)$ from a given transition $(s_t, a_t, s_{t+1}, \phi_t)$ so as to perform the $\xi$-update (13). The first instance is a model-free version and detailed in the following section. A second instance uses a one-step SF model, called One-Step Model-based (MB) SFRQL, which is further described in Appendix B.

**Model-free (MF) SFRQL:** MF SFRQL uses the same principle as standard model-free temporal difference learning methods. The update assumes for a given transition $(s_t, a_t, s_{t+1}, \phi_t)$ that the probability for the observed feature is $p(\phi = \phi_t|s_t, a_t) = 1$. Whereas for all other features $(\forall \phi' \in \Phi, \phi' \neq \phi_t)$ the probability is $p(\phi' = \phi_t|s_t, a_t) = 0$. The resulting updates in case of discrete features are:

$$\phi = \phi_t : \xi^\pi(s_t, a_t, \phi) \leftarrow (1-\alpha)\xi^\pi(s_t, a_t, \phi) + \alpha\Big(1 + \gamma\xi^\pi(s_{t+1}, \bar{a}_{t+1}, \phi)\Big)$$
$$\phi' \neq \phi_t : \xi^\pi(s_t, a_t, \phi') \leftarrow (1-\alpha)\xi^\pi(s_t, a_t, \phi') + \alpha\gamma\xi^\pi(s_{t+1}, \bar{a}_{t+1}, \phi') \,. \tag{16}$$

Appendix D introduces the operator for continuous features. Due to the stochastic update of the $\xi$-function and if the learning rate $\alpha \in (0,1]$ discounts over time, the $\xi$-update will learn the true probability distribution $p(\phi = \phi_t|s_t, a_t)$.

## 3.3 META SFRQ-LEARNING

After discussing SFRQL on a single task and showing its theoretical convergence, we can now investigate how it can be applied in transfer learning. Similar to the linear SF framework the $\xi$-function $\xi^{\pi_i}$ allows to reevaluate a policy learned for task $M_i$ in a new environment $M_j$:

$$Q_j^{\pi_i}(s,a) = \int_\Phi R_j(\phi)\xi^{\pi_i}(s,a,\phi)\mathrm{d}\phi. \tag{17}$$

This allows us to apply GPI in (8) for arbitrary reward functions in a similar manner to what was proposed for linear reward functions in Barreto et al. (2018). We extend the GPI result to SFRQL as follows:

**Theorem 2.** *(Generalised policy improvement in SFRQL) Let $\mathcal{M}$ be the set of tasks, each one associated to a (possibly different) weighting function $R_i \in L^1(\Phi)$. Let $\xi^{\pi_i^*}$ be a representative of the optimal class of $\xi$-functions for task $M_i$, $i \in \{1, \dots, I\}$, and let $\tilde{\xi}^{\pi_i}$ be an approximation to the optimal $\xi$-function, $\|\xi^{\pi_i^*} - \tilde{\xi}^{\pi_i}\|_{R_i} \leq \varepsilon, \forall i$. Then, for another task $M$ with weighting function $R$, the policy defined as:*

$$\pi(s) = \arg\max_a \max_i \int_\Phi R(\phi)\tilde{\xi}^{\pi_i}(s,a,\phi)d\phi, \tag{18}$$

*satisfies:*

$$\|\xi^* - \xi^\pi\|_R \leq \frac{2}{1-\gamma}(\min_i \|R - R_i\|_{p(\phi|s,a)} + \varepsilon), \tag{19}$$

*where $\|f\|_g = \sup_{s,a} \int_\Phi |f \cdot g| \, d\phi$.*

The proof is provided in Appendix A.3.

## 4 EXPERIMENTS

We evaluated SFRQL in two environments[1]. The first has discrete features. It is a modified version of the object collection task by Barreto et al. (2017) having more complex features to allow the usage of general reward functions. See Appendix E.1 for experimental results in the original environment. The second environment, the racer environment, evaluates the agents in tasks with continuous features. We compared different versions of SFRQL, SFQL, and Q-learning which are abbreviated as SFR, SF, and Q respectively. The versions differentiate in if they are learning the feature and reward representations or if these are given.

---

[1]Source code: `https://gitlab.inria.fr/robotlearn/sfr_learning`

## 4.1 Discrete Features - Object Collection Environment

**Environment:** The environment consist of 4 connected rooms (Fig. 1, a). The agent starts an episode in position S and has to learn to reach goal position G. During an episode, the agent collects objects to gain further rewards. Each object has 2 properties: 1) color: orange or blue, and 2) form: box or triangle. The state space is a high-dimensional vector $s \in \mathbb{R}^{112}$. It encodes the agent's position by a $10 \times 10$ grid of two-dimensional Gaussian radial basis functions. Moreover, it includes a memory about which object has been already collected. Agents can move in 4 directions. The features $\phi \in \Phi = \{0, 1\}^5$ are binary vectors. The first 2 dimensions encode if an orange or a blue object was picked up. The 2 following dimensions encode the form. The last dimension encodes if the agent reached goal G. For example, $\phi^\top = [1, 0, 1, 0, 0]$ encodes that the agent picked up an orange box.

**Tasks:** Each agent learns sequentially 300 tasks which differ in their reward for collecting objects. We compared agents in two settings: either in tasks with linear or general reward functions. For each linear task $\mathcal{M}_i$, the rewards $r = \phi^\top \mathbf{w}_i$ are defined by a linear combination of features and a weight vector $\mathbf{w}_i \in \mathbb{R}^5$. The weights $w_{i,k}$ for the first 4 dimensions define the rewards for collecting an object with a specific property. They are randomly sampled from a uniform distribution: $w_{i,k} \sim \mathcal{U}(-1, 1)$. The final weight defines the reward for reaching the goal position which is $w_{i,5} = 1$ for all tasks. The general reward functions are sampled by assigning a different reward to each possible combination of object properties $\phi_j \in \Phi$ using uniform sampling: $R_i(\phi_j) \sim \mathcal{U}(-1, 1)$, such that picking up an orange box might result in a reward of $R_i(\phi^\top = [1, 0, 1, 0, 0]) = 0.23$, whereas a blue box might result in $R_i(\phi^\top = [0, 1, 1, 0, 0]) = 0.76$

**Agents:** We compared SFRQL (SFR) to Q-learning (Q) and classical SF Q-learning (SF) (Barreto et al., 2017). All agents use function approximation for their state-action functions (Q, $\psi$, or $\xi$). An independent mapping is used to map the values from the state for each of the 4 actions.

We evaluated the agents under three conditions. First (Q, SF, SFR), the feature representation $\phi$ and the reward functions are given as defined in the environment and tasks. As the features are discrete, the $\xi$-function is approximated by an independent mapping for each action and possi-

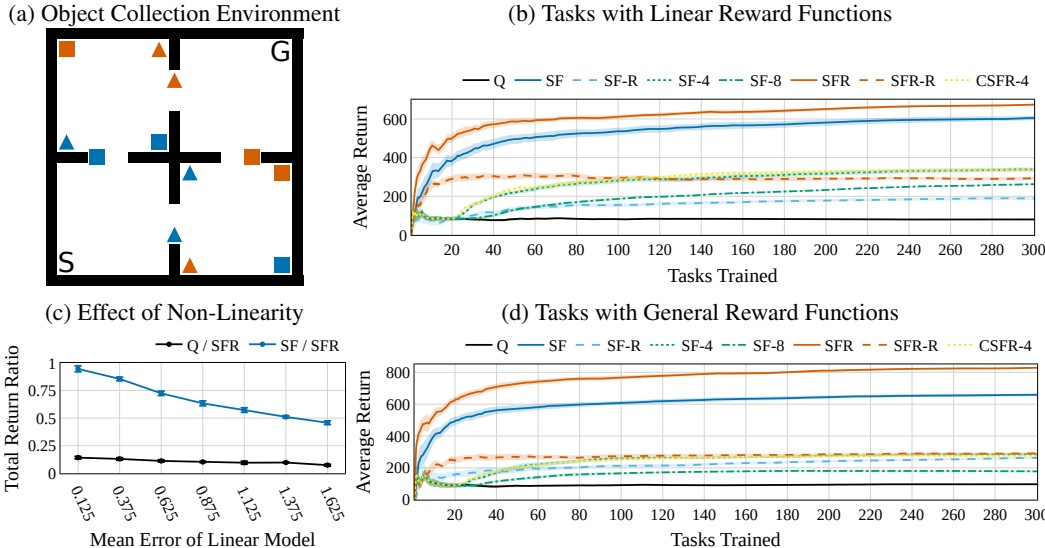

Figure 1: In the object collection environment (a), SFR reached the highest average reward per task for linear (b), and general reward functions (d). The average over 10 runs per algorithm and the standard error of the mean are depicted. (c) The difference between SFR and SF is stronger for general reward tasks that have strong non-linearities, i.e. where a linear reward model yields a high error. SF can only reach less than $50\%$ of SFR's performance in tasks with a mean linear reward model error of $1.625$.

ble feature $\phi \in \Phi$. The Q-value $Q(s, a)$ for the $\xi$-agents (Eq. 12) is computed by: $Q^\pi(s, a) = \sum_{\phi \in \Phi} R(\phi)\xi^\pi(s, a, \phi)$. In tasks with linear reward functions, the sampled reward weights $\mathbf{w}_i$ were given to SF. For general reward functions, SF received an approximated weight vector $\tilde{\mathbf{w}}_i$ based on a linear model that was trained before a task started on several uniformly sampled features and rewards.

In the second condition (SF-R, SFR-R), the feature representation is given, but the reward functions are approximated from observations during the tasks. In detail, the reward weights $\tilde{\mathbf{w}}_i$ were approximated by a linear model $r = \phi^\top \tilde{\mathbf{w}}_i$ for each task.

Under the third condition (SF-$h$, CSFR-$h$), also the feature representation $\tilde{\phi} \in [0, 1]^h$ with $h = 4$ and $h = 8$ was approximated. The features were approximated with a linear model based on observed transitions by QL during the first 20 tasks. The procedure follows the multi-task learning protocol described by (Barreto et al., 2018) based on the work of (Caruana, 1997). As in this case the approximated features are continuous, we introduce a $\xi$-agent for continuous features (CSFR) (Appendix D.2.4). CSFR discretizes each feature dimension $\tilde{\phi}_k \in [0, 1]$ in 11 bins with the bin centers: $X = \{0.0, 0.1, \dots, 1.0\}$. It learns for each dimension $k$ and bin $i$ the $\xi$-value $\xi_k^\pi(s, a, X_i)$. Q-values (Eq. 12) are computed by $Q^\pi(s, a) = \sum_{k=1}^h \sum_{i=1}^{11} r_k(X_i)\xi_k^\pi(s, a, X_i)$. CSFR-8 is omitted as it did not bring a significant advantage.

Each task was executed for $20,000$ steps, and the average performance over 10 runs per algorithm was measured. We performed a grid-search over the parameters of each agent, reporting here the performance of the parameters with the highest total reward over all tasks.

**Results:** SFR outperformed SF and Q for tasks with linear and general reward functions (Fig. 1, b, d). This was the case under the conditions where the features are given and the reward functions were either given (SF, SFR) or had to be learned (SF-R, SFR-R). When features are approximated, both agents (SF-4, CSFR-4) had a similar performance slightly above SFR-R. Approximating 8 features showed a lower performance than 4 features (SF-8, SF-4). We further studied the effect of the strength of the non-linearity in general reward functions on the performance of SFQL compared to SFRQL by evaluating them in tasks with different levels of non-linearity. We sampled general reward functions that resulted in different levels of the mean absolute model error if they are linearly approximated with $\min_{\tilde{\mathbf{w}}} |r(\phi) - \phi^\top \tilde{\mathbf{w}}|$. We trained SF and SFR in each of these conditions on 300 tasks and measured the ratio between the total return of SF to SFR (Fig. 1, c). The relative performance of SF compared to SFR (SF / SFR) decreases with the level of non-linearity. For reward functions that are nearly linear (mean error of $0.125$), both have a similar performance. Whereas, for reward functions that are difficult to model with a linear relation (mean error of $1.625$) SF reaches only less than $50\%$ of the performance of SFR. This follows SF's theoretical limitation in (9) showing the advantage of SFR over SF in non-linear reward tasks.

### 4.2 CONTINUOUS FEATURES - RACER ENVIRONMENT

**Environment and Tasks:** We further evaluated the agents in a 2D environment with continuous features (Fig. 2, a). The agent starts at a random position and drives around for 200 timesteps before the episode ends. Similar to a car, the agent has an orientation and momentum, so that it can only drive straight, or in a right or left curve. The agent reappears on the opposite side if it exits one side. The distance to 3 markers are provided as features $\phi \in \mathbb{R}^3$. Rewards depend on the distances $r = \sum_{k=1}^3 r_k(\phi_k)$, where each component $r_k$ has 1 or 2 preferred distances defined by Gaussian functions. For each of the 40 tasks, the number of Gaussians and their properties ($\mu, \sigma$) are randomly sampled for each feature dimension. Fig. 2 (a) shows a reward function with dark areas depicting higher rewards. The state space is a high-dimensional vector $s \in \mathbb{R}^{120}$ encoding the agent's position and orientation. The 2D position is encoded by a $10 \times 10$ grid of two-dimensional Gaussian radial basis functions. Similarly, the orientation is also encoded using 20 Gaussian radial basis functions.

**Agents:** We used the CSFR agent as the task has continuous feature spaces. The reward weights $\tilde{\mathbf{w}}_i$ for SF were approximated before each task based on randomly sampled features and rewards. We further evaluated agents (SF-R, CSFR-R) that approximated a linear reward model during task execution and agents (SF-$h$, CSFR-$h$) that approximated the features using data collected of the Q-learning agent from the first 10 tasks.

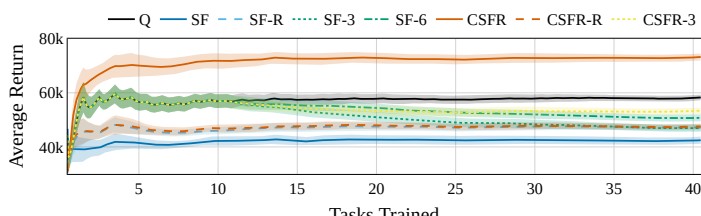

(a) Racer Environment      (b) Tasks with General Reward Functions

Figure 2: (a) Example of a reward function for the racer environment based on distances to its 3 markers. (b) SFRQL (SFR) reaches the highest average reward per task. SF yields a performance even below Q as it is not able to model the reward function with its linear combination of weights and features. The average over 10 runs per agent and the standard error of the mean are depicted.

**Results:** CSFR reached the highest performance of all agents (Fig. 2, b) outperforming Q and SF. SF and all agents that approximated the reward functions (SF-R, CSFR-R) and features (SF-$h$, CSFR-$h$) reached only a performance below Q. They were not able to approximate sufficiently well the Q-function as their reward functions and features depend on a linear reward model which can not represent well the general reward functions in these tasks.

## 5 DISCUSSION AND LIMITATIONS

**SFRQL compared to classical SFQL:** SFRQL allows to disentangle the dynamics of policies in the feature space of a task from the associated reward, see (12). The experimental evaluation in tasks with general reward functions (Fig. 1, d, and Fig. 2) shows that SFRQL can therefore successfully apply GPI to transfer knowledge from learned tasks to new ones. Given a general reward function it can re-evaluate successfully learned policies. Instead, classical SFQL based on a linear decomposition (5) can not be directly applied given a general reward function.

SFRQL also shows an increased performance over SF in environments with linear reward functions (Fig. 1, a). This effect can not be attributed to differences in their computation of a policy's expected return as both are correct (Appendix A.4). Additional experiments showed that SFRQL outperforms SF also in single tasks without transfer. We hypothesis that SFRQL reduces the complexity of the problem for the function approximation compared to a $\psi$-function.

**Learning of Features:** In principle, classical SFQL can also optimize general reward functions if features and reward weights are learned (SF-$h$). This is possible if the learned features describe the non-linear effects in the reward functions. Nonetheless, learning of features adds further challenges and shows a reduced performance in our experiments.

The used feature approximation procedure for all SF-$h$ and SFR-$h$ agents by Barreto et al. (2017) learns features from observations sampled from initial tasks before the GPI procedure starts. Therefore, novel non-linearities potentially introduced at later tasks are not well represent by the learned features. If instead features are learned alongside the GPI procedure, the problem on how to coordinate both learning processes needs to be investigated. Importantly, $\psi$-functions for older tasks would become unusable for the GPI procedure on newer task, because the feature representation changed between them.

Most significantly, our experimental results show that the performance of agents with learned features (SF-$h$) is strongly below the performance of SFRQL with given features (SFR and CSFR). Therefore, if features and reward functions are known then SFRQL outperforms SF, otherwise they are roughly equivalent. Using given features and reward functions is natural for many applications as these are often known, for example in robotic tasks where they are usually manually designed (Akalin & Loutfi, 2021).

**Continuous Feature Spaces:** For tasks with continuous features (racer environment), SFRQL used successfully a discretization of each feature dimension, and learned the $\xi$-values independently for each dimension. This strategy is viable for reward functions that are cumulative over the feature

dimensions: $r(\phi) = \sum_k r_k(\phi_k)$. The Q-value can be computed by summing over the independent dimensions and the bins $X$: $Q^\pi(s,a) = \sum_k \sum_{x \in X} r_k(x)\xi^\pi(s,a,x)$. For more general reward functions, the space of all feature combinations would need to be discretized, which grows exponentially with each new dimension. As a solution the $\xi$-function could be directly defined over the continuous feature space, but this yields some problems. First, the computation of the expected return requires an integral $Q(s,a) = \int_{\phi \in \Phi} R(\phi)\xi(s,a,\phi)$ over features instead of a sum, which is a priori intractable. Second, the representation and training of the $\xi$-function, which would be defined over a continuum thus increasing the difficulty of approximating the function. Janner et al. (2020) and Touati & Ollivier (2021) propose methods that might allow to represent a continuous $\xi$-function, but it is unclear if they converge and if they can be used for transfer learning.

**Computational Complexity:** The improved performance of SFRQL and SFQL over Q-learning comes at the cost of increased computational complexity. The GPI procedure (8) evaluates at each step the $\psi^{\pi_i}$ or $\xi^{\pi_i}$-function over all previous experienced tasks in $\mathcal{M}$. Hence, the computational complexity of both procedures increases linearly with each added task. A solution is to apply GPI only over a subset of learned policies. Nonetheless, how to optimally select this subset is still an open question.

## 6   RELATED WORK

**Transfer Learning:** Transfer methods in RL can be generally categorized according to the type of tasks between which transfer is possible and the type of transferred knowledge (Taylor & Stone, 2009; Lazaric, 2012; Zhu et al., 2020). In the case of SF&GPI which SFRQL is part of, tasks only differ in their reward functions. The type of knowledge that is transferred are policies learned in source tasks which are re-evaluated in the target task and recombined using the GPI procedure. A natural use-case for SFRQL are continual problems (Khetarpal et al., 2020) where an agent has continually adapt to changing tasks, which are in our setting different reward functions.

**Successor Representations & Features:** SFRQL is a variant of SR. SR represent the cumulative probability of future states (Dayan, 1993; White, 1996) whereas SFRQL represents the cumulative probability of low-dimensional future state features. SRs were restricted to low-dimensional state spaces using tabular representations (Wang et al., 2007). Some recent extensions of SRs outside of transfer learning exists, for example, as a basis for exploration (Machado et al., 2020) or active inference (Millidge & Buckley, 2022).

SF extend SR to domains with high-dimensional state spaces (first formulated by Gehring (2015) and then studied in depth in (Kulkarni et al., 2016; Barreto et al., 2017; 2018; Zhang et al., 2017)), by predicting the future occurrence of low-dimensional features that are relevant to define the return. Several extensions to the SF framework have been proposed. One direction aims to learn appropriate features from data such as by optimally reconstructing rewards (Barreto et al., 2017), using the concept of mutual information (Hansen et al., 2019), or grouping of temporal similar states (Madjiheurem & Toni, 2019). Another direction is the generalization of the $\psi$-function over policies (Borsa et al., 2018) analogous to universal value function approximation (Schaul et al., 2015). Similar approaches use successor maps (Madarasz, 2019), goal-conditioned policies (Ma et al., 2020), or successor feature sets (Brantley et al., 2021). Other directions include their application to POMDPs (Vértes & Sahani, 2019), combination with max-entropy principles (Vertes, 2020), or hierarchical RL (Barreto et al., 2021). All these approaches build on the assumption of linear reward functions, whereas SFRQL allows the SF&GPI framework to be used with general reward functions. Nonetheless, most of the extensions for linear SF can be combined with SFRQL.

## 7   CONCLUSION

The introduced SFR framework with its SFRQ-learning algorithm learns the expected cumulative discounted probability of successor features which disentangles the dynamics of a policy in the feature space of a task from the expected rewards. This allows SFRQL to reevaluate the expected return of learned policies for general reward functions and to use it for transfer learning utilizing GPI. We proved that SFRQL converges to the optimal policy, and showed experimentally its improved performance over Q-learning and the classical SF framework for tasks with linear and general reward functions.

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

# A  THEORETICAL RESULTS

This section provides proofs for the converge of SFRQL (Theorem 1) and the performance bound of the GPI procedure (Theorem 2). First it introduces some preliminaries relevant for Theorem 1 before stating the proofs. The final two sections provide more background about the relation between classical SF and SFRQL.

## A.1  PRELIMINARIES

The following Propositions provide the background for the proof the convergence of SFRQL (Section A.2). Depending on the reward function $R$, there are several $\xi$-functions that correspond to the same $Q$ function. Formally, this is an equivalence relationship, and the quotient space has a one-to-one correspondence with the $Q$-function space.

**Proposition 1.** *(Equivalence between functions $\xi$ and Q) Let $R \in L^1(\Phi)$ and $\Xi = \{\xi : \mathcal{S} \times \mathcal{A} \times \Phi \to \mathbb{R}$ s.t. $\xi(s, a, \cdot) \in L^1(\Phi), \forall(s, a) \in \mathcal{S} \times \mathcal{A}, \sup_{s,a} |\int_\Phi R(\phi)\xi(s, a, \phi)d\phi| < \infty\}$ and $\mathcal{Q} = \{Q : \mathcal{S} \times \mathcal{A} \to \mathbb{R}$ s.t. $\|Q\|_\infty < \infty\}$. Let $\sim_R$ be defined as $\xi_1 \sim_R \xi_2 \Leftrightarrow \int_\Phi R(\phi)\xi_1(s, a, \phi)d\phi = \int_\Phi R(\phi)\xi_2(s, a, \phi)d\phi, \forall(s, a) \in \mathcal{S} \times \mathcal{A}$. Then, $\sim_R$ is an equivalence relationship in $\Xi$, and there is a bijective correspondence between the quotient space $\Xi_R$ and $\mathcal{Q}$. The function class corresponding to $\xi$ will be denoted by $[\xi]$.*

*Proof.* We will proof the statements sequentially.

$\sim_R$ **is an equivalence relationship:**    To prove this we need to demonstrate that $\sim_R$ is symmetric, reciprocal and transitive. The three are quite straightforward that $\sim_R$ is defined with an equality.

**Bijective correspondence:**    To prove the bijectivity, we will first prove that it is injective, then surjective. Regarding the injectivity: $[\xi] \neq [\eta] \Rightarrow Q_\xi \neq Q_\eta$, we prove it by contrapositive:

$$Q_\xi = Q_\eta \Rightarrow \int_\Phi R(\phi)\xi(s, a, \phi)\mathrm{d}\phi = \int_\Phi R(\phi)\eta(s, a, \phi)\mathrm{d}\phi \Rightarrow [\xi] = [\eta]. \tag{20}$$

In order to prove the surjectivity, we start from a function $Q \in \mathcal{Q}$ and select an arbitrary $\xi \in \Xi$, then the following function:

$$\xi_Q(s, a, \phi) = \frac{Q(s, a)}{\int_\Phi R(\bar\phi)\xi(s, a, \bar\phi)\mathrm{d}\bar\phi}\xi(s, a, \phi) \tag{21}$$

satisfies that $\xi_Q \in \Xi$ and that $\int_\Phi R(\phi)\xi_Q(s, a, \phi)\mathrm{d}\phi = Q(s, a), \forall(s, a) \in \mathcal{S} \times \mathcal{A}$. We conclude that there is a bijective correspondence between the elements of $\Xi_R$ and of $\mathcal{Q}$. $\qquad \square$

**Corollary 1.** *The bijection between $\Xi_R$ and $\mathcal{Q}$ allows to induce a norm $\|\cdot\|_R$ into $\Xi_R$ from the supremum norm in $\mathcal{Q}$, with which $\Xi_R$ is a Banach space (since $\mathcal{Q}$ is Banach with $\|\cdot\|_\infty$):*

$$\|\xi\|_R = \sup_{s,a} \left| \int_\Phi R(\phi)\xi(s, a, \phi)d\phi \right| = \sup_{s,a} |Q(s, a)| = \|Q\|_\infty . \tag{22}$$

*Proof.* The norm induced in the quotient space is defined from the correspondence between $\Xi_R$ and $\mathcal{Q}$ and is naturally defined as in the previous equation. The norm is well defined since it does not depend on the class representative. Therefore, all the metric properties are transferred, and $\Xi_R$ is immediately Banach with the norm $\|\cdot\|_R$. $\qquad \square$

Similar to the Bellman equation for the Q-function, we can define a Bellman operator for the $\xi$-function, denoted by $T_\xi$, as:

$$T_\xi(\xi^\pi)(s_t, a_t, \phi) = p(\phi|s_t, a_t) + \gamma\mathbb{E}_{p(s_{t+1}|s_t, a_t; \pi)}\{\xi^\pi(s_{t+1}, \bar a_{t+1}, \phi)\}, \tag{23}$$

with $\bar a_{t+1} = \arg\max_a \int_\Phi R(\phi)\xi^\pi(s_{t+1}, a, \phi)\mathrm{d}\phi$. We can use $T_\xi$ to construct a contractive operator:

**Proposition 2.** *(SFRQL has a fixed point) The operator $T_\xi$ is well-defined w.r.t. the equivalence $\sim$, and therefore induces an operator $T_R$ defined over $\Xi_R$. $T_R$ is contractive w.r.t. $\|\cdot\|_R$. Since $\Xi_R$ is Banach, $T_R$ has a unique fixed point and iterating $T_R$ starting anywhere converges to that point.*

*Proof.* We prove the statements above one by one:

**The operator $T_R$ is well defined:** Let us first recall the definition of the operator $T_\xi$ in (23), where we removed the dependency on $\pi$ for simplicity:

$$T_\xi(\xi)(s_t, a_t, \phi) = p(\phi|s_t, a_t) + \gamma\mathbb{E}_{p(s_{t+1}|s_t, a_t)}\left\{\xi(s_{t+1}, \bar{a}_{t+1}, \phi)\right\}.$$

Let $\xi_1, \xi_2 \in [\xi]$ two different representatives of class $[\xi]$, we first observe that the optimal action does not depend on the representative:

$$
\begin{aligned}
\bar{a}_{t+1}^{\xi_1} &= \arg\max_a \int_\Phi R(\phi)\xi_1(s_{t+1}, a, \phi)\mathrm{d}\phi \\
&= \arg\max_a \int_\Phi R(\phi)\xi_2(s_{t+1}, a, \phi)\mathrm{d}\phi = \bar{a}_{t+1}^{\xi_2} =: \bar{a}_{t+1}, \quad \forall s.
\end{aligned}
\tag{24}
$$

We now can write:

$$
\begin{aligned}
&\int_\Phi R(\phi)(T_\xi(\xi_1)(s_t, a_t, \phi) - T_\xi(\xi_2)(s_t, a_t, \phi))\mathrm{d}\phi \\
&= \int_\Phi R(\phi) \int_\mathcal{S} p(s_{t+1}|s_t, a_t)\gamma(\xi_1(s_{t+1}, \bar{a}_{t+1}, \phi) - \xi_2(s_{t+1}, \bar{a}_{t+1}, \phi))\mathrm{d}s_{t+1}\mathrm{d}\phi \\
&= \gamma \int_\mathcal{S} p(s_{t+1}|s_t, a_t) \int_\Phi R(\phi)(\xi_1(s_{t+1}, \bar{a}_{t+1}, \phi) - \xi_2(s_{t+1}, \bar{a}_{t+1}, \phi))\mathrm{d}\phi\mathrm{d}s_{t+1} \\
&= 0
\end{aligned}
\tag{25}
$$

because $\xi_1, \xi_2 \in [\xi]$. Therefore the operator $T_R([\xi]) = T_\xi(\xi)$ is well defined in the quotient space, since the image of class does not depend on the function chosen to represent the class.

**Contractive operator $T_R$:** The contractiveness of $T_R$ can be proven directly:

$$
\begin{aligned}
\|T_R([\xi]) - T_R([\eta])\|_R &= \sup_{s_t, a_t} \left| \int_\Phi R(\phi) \left( p(\phi|s_t, a_t) + \gamma\mathbb{E}_{p(s_{t+1}|s_t, a_t)}\{\xi(s_{t+1}, \bar{a}_{t+1}^\xi, \phi)\} \right. \right. \\
&\qquad\qquad \left. \left. -p(\phi|s_t, a_t) - \gamma\mathbb{E}_{p(s_{t+1}|s_t, a_t)}\{\eta(s_{t+1}, \bar{a}_{t+1}^\eta, \phi)\} \right) \mathrm{d}\phi \right| \\
&= \gamma \sup_{s_t, a_t} \left| \int_\Phi R(\phi)\mathbb{E}_{p(s_{t+1}|s_t, a_t)}\{\xi(s_{t+1}, \bar{a}_{t+1}^\xi, \phi) - \eta(s_{t+1}, \bar{a}_{t+1}^\eta, \phi)\}\mathrm{d}\phi \right| \\
&\leq \gamma \sup_{s_t, a_t} \mathbb{E}_{p(s_{t+1}|s_t, a_t)} \left\{ \left| \sup_{a_{t+1}} \int_\Phi R(\phi)\xi(s_{t+1}, a_{t+1}, \phi)\mathrm{d}\phi \right. \right. \\
&\qquad\qquad \left. \left. - \sup_{a_{t+1}} \int_\Phi R(\phi)\eta(s_{t+1}, a_{t+1}, \phi)\mathrm{d}\phi \right| \right\} \\
&\leq \gamma \sup_{s_{t+1}, a_{t+1}} \left| \int_\Phi R(\phi)(\xi(s_{t+1}, a_{t+1}, \phi) - \eta(s_{t+1}, a_{t+1}, \phi))\mathrm{d}\phi \right| \\
&= \gamma\|[\xi] - [\eta]\|_R
\end{aligned}
\tag{26}
$$

The contractiveness of $T_R$ can also be understood as being inherited from the standard Bellmann operator on $Q$. Indeed, given a $\xi$ function, one can easily see that applying the standard Bellman operator to the $Q$ function corresponding to $\xi$ leads to the $Q$ function corresponding to $T_R([\xi])$.

**Fixed point of $T_R$:** To conclude the proof, we use the fact that any contractive operator on a Banach space, in our case: $T_R : \Xi_R \to \Xi_R$ has a unique fixed point $[\xi^*]$, and that for any starting point $[\xi_0]$, the sequence $[\xi_n] = T_R([\xi_{n-1}])$ converges to $[\xi^*]$ w.r.t. to the corresponding norm $\|[\xi]\|_R$.
$\square$

In other words, successive applications of the operator $T_R$ converge towards the class of optimal $\xi$ functions $[\xi^*]$ or equivalently to an optimal $\xi$ function defined up to an additive function $k$ satisfying $\int_\Phi k(s, a, \phi)R(\phi)\mathrm{d}\phi = 0, \forall(s, a) \in \mathcal{S} \times \mathcal{A}$ (i.e. $k \in \mathrm{Ker}(\xi \to \int_\Phi R\xi)$).

While these two results (Propositions 1 and 2) state the theoretical links to standard Q-learning formulations, the $T_\xi$ operator defined in (23) is not usable in practice, because of the expectation.

## A.2 PROOF OF THEOREM 1

Propositions 1 and 2 are useful to prove that the $\xi$ learning iterates converge in $\Xi_R$. Let us restate the definition of the operator from (13):

$$\xi_{k+1}^\pi(s_t, a_t, \phi) \leftarrow \xi_k^\pi(s_t, a_t, \phi) + \alpha_k \left[ p(\phi_t = \phi | s_t, a_t; \pi) + \gamma \xi_k^\pi(s_{t+1}, \bar{a}_{t+1}, \phi) - \xi_k^\pi(s_t, a_t, \phi) \right]$$

and the theoretical result:

**Theorem 1.** *(Convergence of SFRQL) For a sequence of state-action-feature $\{s_t, a_t, s_{t+1}, \phi_t\}_{t=0}^\infty$ consider the SFRQL update given in (13). If the sequence of state-action-feature triples visits each state, action infinitely often, and if the learning rate $\alpha_k$ is an adapted sequence satisfying the Robbins-Monro conditions:*

$$\sum_{k=1}^\infty \alpha_k = \infty, \qquad \sum_{k=1}^\infty \alpha_k^2 < \infty \tag{27}$$

*then the sequence of function classes corresponding to the iterates converges to the optimum, which corresponds to the optimal Q-function to which standard Q-learning updates would converge to:*

$$[\xi_n] \to [\xi^*] \quad \text{with} \quad Q^*(s, a) = \int_\Phi R(\phi) \xi^*(s, a, x) d\phi. \tag{28}$$

*Proof.* The proof re-uses the flow of the proof used for Q-learning (Tsitsiklis, 1994). Indeed, we rewrite the operator above as:

$$\xi_{k+1}^\pi(s_t, a_t, \phi) \leftarrow \xi_k^\pi(s_t, a_t, \phi) + \alpha_k \left[ T_\xi(\xi_k^\pi)(s_t, a_t, \phi) - \xi_k^\pi(s_t, a_t, \phi) + \varepsilon(s_t, a_t, \phi) \right]$$

with $\varepsilon$ defined as:

$$\varepsilon(s_t, a_t, \phi) = \begin{aligned}&p(\phi_t = \phi | s_t, a_t; \pi) + \gamma \xi_k^\pi(s_{t+1}, \bar{a}_{t+1}, \phi) \\ &- \mathbb{E}\left\{ p(\phi_t = \phi | s_t, a_t; \pi) + \gamma \xi_k^\pi(s_{t+1}, \bar{a}_{t+1}, \phi) \right\}.\end{aligned}$$

Obviously $\varepsilon$ satisfies $\mathbb{E}\{\varepsilon\} = 0$, which, together with the contractiveness of $T_R$, is sufficient to demonstrate the convergence of the iterative procedure as done for Q-learning. In our case, the optimal function $\xi^*$ is defined up to an additive kernel function $\kappa \in \text{Ker}\left(\xi \to \int_\Phi R\xi\right)$. The correspondence with the optimal Q learning function is a direct application of the correspondence between the $\xi$- and Q-learning problems. $\qquad \square$

Generally speaking, the exact probability distribution of the features $p(\phi_t = \phi | s_t, a_t; \pi)$ is not available. In our model-free algorithm, this distribution is replaced by the indicator function of the observed feature, whose expectation is equal to the exact distribution, and therefore the condition $\mathbb{E}\{\varepsilon\} = 0$ is satisfied. In our model-based algorithm proposed below, we need to be sure that the estimated distribution is not biased, so that the zero-mean condition is also satisfied.

## A.3 PROOF OF THEOREM 2

Let us restate the result.

**Theorem 2.** *(Generalised policy improvement in SFRQL) Let $\mathcal{M}$ be the set of tasks, each one associated to a (possibly different) weighting function $R_i \in L^1(\Phi)$. Let $\xi^{\pi_i^*}$ be a representative of the optimal class of $\xi$-functions for task $M_i$, $i \in \{1, \ldots, I\}$, and let $\tilde{\xi}^{\pi_i}$ be an approximation to the optimal $\xi$-function, $\|\xi^{\pi_i^*} - \tilde{\xi}^{\pi_i}\|_{R_i} \leq \varepsilon, \forall i$. Then, for another task $M$ with weighting function $R$, the policy defined as:*

$$\pi(s) = \arg\max_a \max_i \int_\Phi R(\phi) \tilde{\xi}^{\pi_i}(s, a, \phi) d\phi, \tag{29}$$

*satisfies:*

$$\|\xi^* - \xi^\pi\|_R \leq \frac{2}{1-\gamma} (\min_i \|R - R_i\|_{p(\phi | s, a)} + \varepsilon), \tag{30}$$

*where $\|f\|_g = \sup_{s,a} \int_\Phi |f \cdot g| \, d\phi$.*

*Proof.* The proof is stated in two steps. First, we exploit the proof of Proposition 1 of (Barreto et al., 2017), and in particular (13) that states:

$$\|Q^* - Q^\pi\|_\infty \leq \frac{2}{1-\gamma} \left( \sup_{s,a} |r(s,a) - r_i(s,a)| + \varepsilon \right), \quad \forall i \in \{1, \ldots, I\}, \tag{31}$$

where $Q^*$ and $Q^\pi$ are the Q-functions associated to the optimal and $\pi$ policies in the environment $R$. The conditions on the Q functions required in the original proposition are satisfied because the $\xi$-functions satisfy them, and there is an isometry between Q and $\xi$ functions.

Because the above inequality is true for all training environments $i$, we can rewrite as:

$$\|Q^* - Q^\pi\|_\infty \leq \frac{2}{1-\gamma} \left( \min_i \sup_{s,a} |r(s,a) - r_i(s,a)| + \varepsilon \right). \tag{32}$$

We now realise that, in the case of SFRQL, the reward functions rewrite as:

$$r(s,a) = \int_\Phi R(\phi) p(\phi|s,a) \mathrm{d}\phi \qquad r_i(s,a) = \int_\Phi R_i(\phi) p(\phi|s,a) \mathrm{d}\phi, \tag{33}$$

and therefore we have:

$$\begin{aligned}
\sup_{s,a} |r(s,a) - r_i(s,a)| &= \sup_{s,a} \left| \int_\Phi (R(\phi) - R_i(\phi)) p(\phi|s,a) \mathrm{d}\phi \right| \\
&\leq \sup_{s,a} \int_\Phi |R(\phi) - R_i(\phi)| \, p(\phi|s,a) \mathrm{d}\phi \\
&= \|R - R_i\|_{p(\phi|s,a)}
\end{aligned} \tag{34}$$

Similarly, due to the isometry between $\xi$ and Q-learning, i.e. Proposition 2, we can write that:

$$\begin{aligned}
\|\xi^* - \xi^\pi\|_R = \|[\xi^*] - [\xi^\pi]\|_R &= \|Q^* - Q^\pi\|_\infty \\
&\leq \frac{2}{1-\gamma} \left( \min_i \sup_{s,a} |r(s,a) - r_i(s,a)| + \varepsilon \right) \\
&\leq \frac{2}{1-\gamma} (\min_i \|R - R_i\|_{p(\phi|s,a)} + \varepsilon),
\end{aligned} \tag{35}$$

which proves the generalised policy improvement for SFRQL. $\square$

### A.4 RELATION BETWEEN CLASSICAL SFQL AND SFRQL FOR LINEAR REWARD FUNCTIONS

In the case of linear reward functions, i.e. where assumption (5) holds, it is possible to show that SFRQL can be reduced to classical SF. Classical SF represents therefore a specific case of SFRQL under this assumption.

**Theorem 3.** *(Equality of classical SF and SFRQL for linear reward functions) Given the assumption that reward functions are linearily decomposable with*

$$r_i(s_t, a_t, s_{t+1}) \equiv \phi(s_t, a_t, s_{t+1})^\top \mathbf{w}_i \,,$$

*where $\phi \in \mathbb{R}^n$ are features for a transition and $\mathbf{w}_i \in \mathbb{R}^n$ are the reward weight vector of task $m_i \in \mathcal{M}$, then the classical SF and SFRQL framework are equivalent.*

*Proof.* We start with the definition of the Q-value according to SFRQL from (12). After replacing the reward function $R_i$ with our linear assumption, the definition of the Q-function according to

classical SF with the $\psi$-function can be recovered:

$$
\begin{aligned}
Q_i(s,a) &= \int_\Phi \xi^\pi(s_t, a_t, \phi) R_i(\phi) \mathrm{d}\phi \\
&= \int_\Phi \xi^\pi(s_t, a_t, \phi) \phi^\top \mathbf{w}_i \mathrm{d}\phi \\
&= \mathbf{w}_i^\top \int_\Phi \sum_{k=0}^\infty \gamma^k p(\phi_{t+k} = \phi | s_t = s, a_t = a; \pi) \phi \; \mathrm{d}\phi \\
&= \mathbf{w}_i^\top \sum_{k=0}^\infty \gamma^k \int_\Phi p(\phi_{t+k} = \phi | s_t = s, a_t = a; \pi) \phi \; \mathrm{d}\phi \\
&= \mathbf{w}_i^\top \sum_{k=0}^\infty \gamma^k \mathbb{E} \{\phi_{t+k}\} \\
&= \mathbb{E} \left\{ \sum_{k=0}^\infty \gamma^k \phi_{t+k} \right\}^\top \mathbf{w}_i \quad = \quad \psi(s,a)^\top \mathbf{w}_i \; .
\end{aligned}
$$

$\square$

Please note, although both methods are equal in terms of their computed values, how these are represented and learned differs between them. Thus, it is possible to see a performance difference of the methods in the experimental results where SFRQL outperforms SF in our environments.

## A.5  RELATION BETWEEN CLASSICAL SF AND SFRQL FOR GENERAL REWARD FUNCTIONS

An inherent connection between classical SF and SFRQL exists for general reward functions that have either a discrete feature space or where the feature space is discretized. In these cases SFRQL can be viewed as a reformulation of the original feature space and general reward functions into a feature space with linear reward functions which are then solved with the classical SF mechanism. Nonetheless, in continuous feature spaces, such a reformulation is not possible. SFRQL represents in this case a new algorithm that is able to converge to the optimal policy (Theorem 1) in difference to the classical SF method. The following section analyzes each of the three cases.

**Discrete Feature Space**  Given an ordered set of discrete features $\phi \in \{\phi_1, \ldots, \phi_j, \ldots, \phi_m\} = \Phi$ with $m = |\Phi|$ and general reward functions $r_i(\phi) \in \mathbb{R}$. We can reformulate this problem into a linear decomposable problem using binary features $\varphi \in \{0,1\}^m$ and a reward weight vector $\mathbf{w}_i \in R^m$. The feature vector $\varphi$ represents a binary pointer, where its $j$'th element ($\varphi_j$) is 1 if the observed feature $\phi_t$ is the $j$'th feature ($\phi_j$) of the original feature set $\Phi$:

$$
\varphi_j = \begin{cases} 1 & | \; \phi_t = \phi_j \\ 0 & | \; \text{otherwise} \end{cases} . \tag{36}
$$

The $j$'th element in the reward weight vector $\bar{w}$ corresponds to the reward of the $j$'th feature in $\Phi$: $w_{i,j} = r_i(\phi_j)$. Given such a reformulation, we can see that it is possible to reformulate the reward function using a linear composition: $r_i(\phi) = \varphi^\top \mathbf{w}_i$. Based on the reformulation, we can define the revised SF function:

$$
\psi_\varphi(s,a) = \mathbb{E}_\pi \left\{ \sum_{k=0}^\infty \gamma^k \varphi_{t+k} \, | s_t = s, a_t = a \right\} . \tag{37}
$$

If we concentrate on a single element $j$ of this vector function and replace the expectation by the probabilities of the feature element being 0 or 1 at a certain time point, then we recover SFRQL:

$$
\begin{aligned}
\psi_\varphi(s,a)_j &= \mathbb{E}_\pi \left\{ \sum_{k=0}^\infty \gamma^k \varphi_{j,t+k} \,|\, s_t = s, a_t = a \right\} \\
&= \sum_{k=0}^\infty \gamma^k \mathbb{E}_\pi \left\{ \varphi_{j,t+k} \,|\, s_t = s, a_t = a \right\} \\
&= \sum_{k=0}^\infty \gamma^k p(\varphi_{j,t+k} = 0 | s_t = s, a_t = a; \pi) \cdot 0 + p(\varphi_{j,t+k} | s_t = s, a_t = a; \pi) \cdot 1 \quad (38) \\
&= \sum_{k=0}^\infty \gamma^k p(\varphi_{j,t+k} = 1 | s_t = s, a_t = a; \pi) \\
&= \sum_{k=0}^\infty \gamma^k p(\phi_{t+k} = \phi_j | s_t = s, a_t = a; \pi) \;=\; \xi(s, a, \phi_j) .
\end{aligned}
$$

In conclusion, this shows the underlying principle of how SFRQL solves problems with discrete feature spaces. It reformulates the feature space and the reward functions into a form that can then be solved via a linear composition.

**Discretized Feature Space** Discretized continuous feature spaces as used by the CSFR agent can also be reformulated to a linear decomposable problem. The proof follows the same logic as the proof for discrete feature spaces. In this case binary index features for each bin of the discretized feature dimension of the CSFR can be constructed.

**Continuous Feature Space** In the case of continuous feature spaces $\Phi = \mathbb{R}^n$, a reconstruction of the feature space and the reward functions to construct a problem with a linear composition of rewards similar to the one given for discrete feature spaces is not possible. In this case, it is not possible to define a vector whos element point to each possible element in $\Phi$ as $\Phi \in \mathbb{R}^n$ is a uncountable set. Therefore, SFRQL can not be recovered by reformulating the feature space and the reward functions. It represents a class of algorithms that can not be reduced to classical SF. Nonetheless, the convergence proof of SFRQL also holds under this condition, whereas classical SF can not solve such environments.

## B ONE-STEP SF MODEL-BASED (MB) SFRQL

Besides the MF SFRQL update operator (16), we introduce a second SFRQL procedure called One-step SF Model-based (MB) SFRQL that attempts to reduce the variance of the update. To do so, MB SFRQL estimates the distribution over the successor features over time. Let $\tilde{p}(\phi_t = \phi | s_t, a_t; \pi)$ denote the current estimate of the feature distribution. Given a transition $(s_t, a_t, s_{t+1}, \phi_t)$ the model is updated according to:

$$
\begin{aligned}
\phi = \phi_t : \quad & \tilde{p}_\phi(\phi | s_t, a_t; \pi) \;\leftarrow\; \tilde{p}_\phi(\phi | s_t, a_t; \pi) + \beta\left(1 - \tilde{p}_\phi(\phi | s_t, a_t; \pi)\right) \\
\phi' \neq \phi_t : \quad & \tilde{p}_\phi(\phi' | s_t, a_t; \pi) \;\leftarrow\; \tilde{p}_\phi(\phi' | s_t, a_t; \pi) - \beta \tilde{p}_\phi(\phi' | s_t, a_t; \pi) ,
\end{aligned}
$$

where $\beta \in [0,1]$ is the learning rate. After updating the model $\tilde{p}_\phi$, it can be used for the $\xi$-update as defined in (13). Since the learned model $\tilde{p}_\phi$ is independent from the reward function and from the policy, it can be learned and used over all tasks.

## C EXPERIMENTAL DETAILS: OBJECT COLLECTION ENVIRONMENT

The object collection environment (Fig. 1, a) was briefly introduced in Section 4.1. This section provides a formal description.

## C.1    ENVIRONMENT

The environment is a continuous two-dimensional area in which the agent moves. The position of the agent is a point in the 2D space: $(x, y) \in [0, 1]^2$. The action space of the agent consists of four movement directions: $A = \{\text{up, down, left, right}\}$. Each action changes the position of the agent in a certain direction and is stochastic by adding a Gaussian noise. For example, the action for going right updates the position according to $x_{t+1} = x_t + \mathcal{N}(\mu = 0.05, \sigma = 0.005)$. If the new position ends in a wall (black areas in Fig. 1, a) that have a width of 0.04) or outside the environment, the agent is set back to its current position. Each environment has 12 objects. Each object has two properties with two possible values: color (orange, blue) and shape (box, triangle). If the agent reaches an object, it collects the object which then disappears. The objects occupy a circular area with radius 0.04. At the beginning of an episode the agent starts at location S with $(x, y)_{\text{S}} = (0.05, 0.05)$. An episode ends if the agent reaches the goal area G which is at position $(x, y)_{\text{G}} = (0.86, 0.86)$ and has a circular shape with radius 0.1. After an episode the agent is reset to the start position S and all collected objects reappear.

The state space of the agents consist of their position in the environment and the information about which objects they already collected during an episode. Following (Barreto et al., 2017), the position is encoded using a radial basis function approach. This upscales the agent's $(x, y)$ position to a high-dimensional vector $s_{\text{pos}} \in \mathbb{R}^{100}$ providing a better signal for the function approximation of the different functions such as the $\psi$ or $\xi$-function. The vector $s_{\text{pos}}$ is composed of the activation of two-dimensional Gaussian functions based on the agents position $(x, y)$:

$$s_{\text{pos}} = \exp\left(-\frac{(x - c_{j,1})^2 + (y - c_{j,2})^2}{\sigma}\right), \tag{39}$$

where $c_j \in \mathbb{R}^2$ is the center of the $j^{\text{th}}$ Gaussian. The centers are laid out on a regular $10 \times 10$ grid over the area of the environment. The state also encodes the memory about the objects that the agent has already collected using a binary vector $s_{\text{mem}} \in \{0, 1\}^{12}$. The $j^{\text{th}}$ dimension encodes if the $j^{\text{th}}$ object has been taken ($s_{\text{mem},j} = 1$) or not ($s_{\text{mem},j} = 0$). An additional constant term was added to the state to aid the function approximation. As a result, the state received by the agents is a column vector with $s = [s_{\text{pos}}^\top, s_{\text{mem}}^\top, 1]^\top \in \mathbb{R}^{113}$.

The features $\phi(s_t, a_t, s_{t+1}) \in \Phi \subset \{0, 1\}^5$ in the environment describe the type of object that was collected by an agent during a step or if it reached the goal position. The first four feature dimensions encode binary the properties of a collected object and the last dimension if the goal area was reached. In total $|\Phi| = 6$ possible features exists: $\phi_1 = [0, 0, 0, 0, 0]^\top$ - standard observation, $\phi_2 = [1, 0, 1, 0, 0]^\top$ - collected an orange box, $\phi_3 = [1, 0, 0, 1, 0]^\top$ - collected an orange triangle, $\phi_4 = [0, 1, 1, 0, 0]^\top$ - collected a blue box, $\phi_5 = [0, 1, 0, 1, 0]^\top$ - collected a blue triangle, and $\phi_6 = [0, 0, 0, 0, 1]^\top$ - reached the goal area.

Two types of tasks were evaluated in this environment that have either 1) linear or 2) general reward functions. 300 tasks, i.e. reward functions, were sampled for each type. For linear tasks, the rewards $r = \phi^\top \mathbf{w}_i$ are defined by a linear combination of discrete features $\phi \in \mathbb{N}^5$ and a weight vector $\mathbf{w}_i \in \mathbb{R}^5$. The first four dimensions in $\mathbf{w}_i$ define the reward that the agent receives for collecting objects having specific properties, e.g. being blue or being a box. The weights for each of the four dimensions are randomly sampled from a uniform distribution: $\mathbf{w}_{k \in [1,2,3,4]} \sim \mathcal{U}(-1, 1)$ for each task. The final weight defines the reward for reaching the goal state which is $\mathbf{w}_5 = 1$ for each task. For training agents in general reward tasks, general reward functions $R_i$ for each task $M_i$ were sampled. These reward functions define for each of the four features ($\phi_2, \ldots, \phi_5$) that represent the collection of a specific object type an individual reward. Their rewards were sampled from a uniform distribution: $R_i(\phi_{k \in \{2,\ldots,5\}}) \sim \mathcal{U}(-1, 1)$. The reward for collecting no object is $R_i(\phi_1) = 0$ and for reaching the goal area is $R_i(\phi_6) = 1$ for all tasks. Reward functions of this form can not be linearly decomposed in features and a weight vector.

## C.2    ALGORITHMS

This section introduces the details of each evaluated algorithm. First the common elements are discussed before introducing their specific implementations.

All agents experience the tasks $M \in \mathcal{M}$ of an environment sequentially. They are informed when a new task starts. All algorithms receive the features $\phi(s, a, s')$ of the environment. For the action

selection and exploration, all agents use a $\epsilon$-greedy strategy. With probability $\epsilon \in [0, 1]$ the agent performs a random action. Otherwise it selects the action that maximizes the expected return.

As the state space ($s \in \mathbb{R}^{113}$) of the environments is high-dimensional and continuous, all agents use an approximation of their respective functions such as for the Q-function ($\tilde{Q}(s, a) \approx Q(s, a)$) or the $\xi$-function ($\tilde{\xi}(s, a, \phi) \approx \xi(s, a, \phi)$). We describe the general function approximation procedure on the example of $\xi$-functions. If not otherwise mentioned, all functions are approximated by a single linear mapping from the states to the function values. The parameters $\theta^\xi \in \mathbb{R}^{113 \times |\mathcal{A}| \times |\Phi|}$ of the mapping have independent components $\theta^\xi_{a,\phi} \in \mathbb{R}^{113}$ for each action $a \in \mathcal{A}$ and feature $\phi \in \Phi$:

$$\tilde{\xi}(s, a, \phi; \theta^\xi) = s^\top \theta^\xi_{a,\phi} \tag{40}$$

To learn $\tilde{\xi}$ we update the parameters $\theta^\xi$ using stochastic gradient descent following the gradients $\nabla_{\theta^\xi} \mathcal{L}_\xi(\theta^\xi)$ of the loss based on the SFRQL update (13):

$$\forall \phi \in \Phi: \ \mathcal{L}_\xi(\theta^\xi) = \mathbb{E}\left\{\left(p(\phi_t = \phi | s_t, a_t) + \gamma \tilde{\xi}(s_{t+1}, \bar{a}_{t+1}, \phi; \bar{\theta}^\xi) - \tilde{\xi}(s_t, a_t, \phi; \theta^\xi)\right)^2\right\}$$
$$\text{with } \bar{a}_{t+1} = \arg\max_a \sum_{\phi \in \Phi} R(\phi)\tilde{\xi}(s_{t+1}, a, \phi; \bar{\theta}^\xi), \tag{41}$$

where $\bar{\theta}^\xi = \theta^\xi$ but $\bar{\theta}^\xi$ is treated as a constant for the purpose of calculating the gradients $\nabla_{\theta^\xi} \mathcal{L}_\xi(\theta^\xi)$. We used PyTorch[2] for the computation of gradients and its stochastic gradient decent procedure (SGD) for updating the parameters.

### C.2.1   LEARNING OF FEATURES REPRESENTATIONS

Besides the usage of the predefined environment features $\Phi$ (Appendix C.1), the algorithms have been also evaluated with learned features $\tilde{\phi}_h \in \Phi_h \subset \mathbb{R}^h$. Learned features are represented by a linear mapping: $\tilde{\phi}_h(s_t, s_{t+1}) = \varsigma(c(s_t, s_{t+1})^\top H$. $\varsigma(x) = \frac{1}{1+\exp(-x)}$ is an element-wise applied sigmoid function. The input is a concatenation of the current and next observation of the transition $c(s_t, s_{t+1}) \in \mathbb{R}^{226}$. The learnable parameters $H \in \mathbb{R}^{226 \times h}$ define the linear mapping.

The learning procedure for $H$ follows the multi-task framework described in (Barreto et al., 2017) based on (Caruana, 1997). $H$ is learned to optimize the representation of the reward function $r = \phi_h^\top \mathbf{w}_i$ over several tasks. As the reward weights $\mathbf{w}_i$ of a task $M_i$ are not known for learned features, they also need to be learned per task. Both parameters ($H, \mathbf{w}$) were learned on a dataset of transitions by the QL algorithm (Section C.2.2) during the initial 20 tasks of the object collection environment. Many of the transitions have a reward of 0 which would produce a strong bias to learn features that always predict a zero reward. Therefore, 75% of zero reward transitions have been filtered out. Based on the collected dataset the parameters $H$, and $\mathbf{w}_{i \in 1, \ldots 20}$ were learned using a gradient decent optimization based on the following mean squared error loss:

$$\mathcal{L}_H(H, \mathbf{w}_i) = \mathbb{E}_{(s_t, s_{t+1}, r_t) \sim \mathcal{D}'_i}\left\{\left(\varsigma(c(s_t, s_{t+1})^\top H)^\top \mathbf{w}_i - r_t\right)^2\right\}$$

where $\mathcal{D}'_i$ is the dataset of transitions from which the zero reward transitions were removed.

As each algorithm was evaluated for 10 runs (Section C.3), we also learned for each run features from the dataset of transitions from the QL algorithm of the same run. The parameters $H$ and $\mathbf{w}_{i \in 1, \ldots 20}$ were optimized with Adam (learning rate of 0.003). They were optimized for $1, 000, 000$ iterations. In each iteration, a random batch of 128 transitions over all 20 tasks in the dataset was used. $H$ and each $\mathbf{w}_i$ were initialized using a normal distribution with $\sigma = 0.05$. Features $\phi_h$ with dimensions $h = 4$ and $h = 8$ were learned.

### C.2.2   QL

The Q-learning (QL) agent (Algorithm 1) represents standard Q-learning (Watkins & Dayan, 1992). The Q-function is approximated and updated using the following loss after each observed transition:

$$\mathcal{L}_Q(\theta^Q) = \mathbb{E}\left\{\left(r(s_t, a_t, s_{t+1}) + \gamma \max_{a_{t+1}} \tilde{Q}(s_{t+1}, a_{t+1}; \bar{\theta}^Q) - \tilde{Q}(s_t, a_t; \theta^Q)\right)^2\right\} \tag{42}$$

---

[2]PyTorch v1.4: https://pytorch.org

where $\bar{\theta}^Q = \theta^Q$ but $\bar{\theta}^Q$ is treated as a constant for the purpose of optimization, i.e no gradients flow through it. Following (Barreto et al., 2017) the parameters $\theta^Q$ are reinitialized for each new task.

---

**Algorithm 1:** Q-learning (QL)

---

**Input:** exploration rate: $\epsilon$
learning rate for the Q-function: $\alpha$

**for** $i \leftarrow 1$ **to** *num_tasks* **do**
    initialize $\tilde{Q}$: $\theta^Q \leftarrow$ small random initial values
    new_episode $\leftarrow$ true
    **for** $t \leftarrow 1$ **to** num_steps **do**
        **if** *new_episode* **then**
            new_episode $\leftarrow$ false
            $s_t \leftarrow$ initial state
        With probability $\epsilon$ select a random action $a_t$, otherwise $a_t \leftarrow \arg\max_a \tilde{Q}(s_t, a)$
        Take action $a_t$ and observe reward $r_t$ and next state $s_{t+1}$
        **if** $s_{t+1}$ *is a terminal state* **then**
            new_episode $\leftarrow$ true
            $\gamma_t \leftarrow 0$
        **else**
            $\gamma_t \leftarrow \gamma$
        $y \leftarrow r_t + \gamma_t \max_{a_{t+1}} \tilde{Q}(s_{t+1}, a_{t+1})$
        Update $\theta^Q$ using SGD($\alpha$) with $\mathcal{L}_Q = (y - \tilde{Q}(s_t, a_t))^2$
        $s_t \leftarrow s_{t+1}$

---

### C.2.3 SF

The classical successor feature algorithm (SF) is based on a linear decomposition of rewards in features and reward weights (Barreto et al., 2017) (Algorithm 2). If the agent is learning the reward weights $\tilde{\mathbf{w}}_i$ for a task $M_i$ then they are randomly initialized at the beginning of a task. For the case of general reward functions and where the reward weights are given to the agents, the weights are learned to approximate a linear reward function before the task. See Section C.3 for a description of the training procedure. After each transition the weights are updated by minimizing the error between the predicted rewards $\phi(s_t, a_t, s_{t+1})^\top \tilde{\mathbf{w}}_i$ and the observed reward $r_t$:

$$\mathcal{L}_{\mathbf{w}_i}(\tilde{\mathbf{w}}_i) = \mathbb{E}\left\{ \left( r(s, a, s') - \phi(s_t, a_t, s_{t+1})^\top \tilde{\mathbf{w}}_i \right)^2 \right\} . \tag{43}$$

SF learns an approximated $\tilde{\psi}_i$-function for each task $M_i$. The parameters of the $\tilde{\psi}$-function for the first task $\theta_1^\psi$ are randomly initialized. For consecutive tasks, they are initialized by copying them from the previous task ($\theta_i^\psi \leftarrow \theta_{i-1}^\psi$). The $\tilde{\psi}_i$-function of the current task $M_i$ is updated after each observed transition with the loss based on (7):

$$\mathcal{L}_\psi(\theta_i^\psi) = \mathbb{E}\left\{ \left( \phi(s_t, a_t, s_{t+1}) + \gamma\tilde{\psi}_i(s_{t+1}, \bar{a}_{t+1}; \bar{\theta}_i^\psi) - \tilde{\psi}_i(s_t, a_t; \theta_i^\psi) \right)^2 \right\}$$
$$\text{with } \bar{a}_{t+1} = \arg\max_a \max_{k \in \{1,2,\dots,i\}} \tilde{\psi}_k(s_{t+1}, a; \bar{\theta}_k^\psi)^\top \tilde{\mathbf{w}}_i , \tag{44}$$

where $\bar{\theta}_i^\psi = \theta_i^\psi$ but $\bar{\theta}_i^\psi$ is treated as a constant for the purpose of optimization, i.e no gradients flow through it. Besides the current $\tilde{\psi}_i$-function, SF also updates the $\tilde{\psi}_c$-function which provided the GPI optimal action for the current transition: $c = \arg\max_{k \in \{1,2,\dots,i\}} \max_b \tilde{\psi}_k(s, b)^\top \tilde{\mathbf{w}}_i$. The update uses the same loss as for the update of the active $\tilde{\psi}_i$-function (44), but instead of using the GPI optimal action as next action, it uses the optimal action according to its own policy: $\bar{a}_{t+1} = \arg\max_a \tilde{\psi}_c(s_{t+1}, a)^\top \tilde{\mathbf{w}}_c$

---

**Algorithm 2:** Classical SF Q-learning (SF) (Barreto et al., 2017)

---

**Input:** exploration rate: $\epsilon$
       learning rate for $\psi$-functions: $\alpha$
       learning rate for reward weights $\mathbf{w}$: $\alpha_{\mathbf{w}}$
       features $\phi$ or $\tilde{\phi}$
       optional: reward weights for tasks: $\{\tilde{\mathbf{w}}_1, \tilde{\mathbf{w}}_2, \ldots, \tilde{\mathbf{w}}_{\text{num\_tasks}}\}$

---

**for** $i \leftarrow 1$ **to** *num\_tasks* **do**
  **if** $\tilde{\mathbf{w}}_i$ *not provided* **then** $\tilde{\mathbf{w}}_i \leftarrow$ small random initial values
  **if** $i = 1$ **then** initialize $\tilde{\psi}_i$: $\theta_i^{\psi} \leftarrow$ small random initial values **else** $\theta_i^{\psi} \leftarrow \theta_{i-1}^{\psi}$
  new\_episode $\leftarrow$ true
  **for** $t \leftarrow 1$ **to** num\_steps **do**
    **if** *new\_episode* **then**
      new\_episode $\leftarrow$ false
      $s_t \leftarrow$ initial state
    $c \leftarrow \arg\max_{k \in \{1,2,\ldots,i\}} \max_a \tilde{\psi}_k(s_t, a)^{\top} \tilde{\mathbf{w}}_i$       // GPI optimal policy
    With probability $\epsilon$ select a random action $a_t$, otherwise $a_t \leftarrow \arg\max_a \tilde{\psi}_c(s_t, a)^{\top} \tilde{\mathbf{w}}_i$
    Take action $a_t$ and observe reward $r_t$ and next state $s_{t+1}$
    Update $\tilde{\mathbf{w}}_i$ using SGD($\alpha_{\mathbf{w}}$) with $\mathcal{L}_{\mathbf{w}} = (r_t - \phi(s_t, a_t, s_{t+1})^{\top} \tilde{\mathbf{w}}_i)^2$
    **if** $s_{t+1}$ *is a terminal state* **then**
      new\_episode $\leftarrow$ true
      $\gamma_t \leftarrow 0$
    **else**
      $\gamma_t \leftarrow \gamma$
    // GPI optimal next action for task $i$
    $\bar{a}_{t+1} \leftarrow \arg\max_a \arg_{k \in \{1,2,\ldots,i\}} \tilde{\psi}_k(s_{t+1}, a)^{\top} \tilde{\mathbf{w}}_i$
    $y \leftarrow \phi(s_t, a_t, s_{t+1}) + \gamma_t \tilde{\psi}_i(s_{t+1}, \bar{a}_{t+1})$
    Update $\theta_i^{\psi}$ using SGD($\alpha$) with $\mathcal{L}_{\psi} = (y - \tilde{\psi}_i(s_t, a_t))^2$
    **if** $c \neq i$ **then**
      $\bar{a}_{t+1} \leftarrow \arg\max_a \tilde{\psi}_c(s_{t+1}, a)^{\top} \tilde{\mathbf{w}}_c$  // optimal next action for task $c$
      $y \leftarrow \phi(s_t, a_t, s_{t+1}) + \gamma_t \tilde{\psi}_c(s_{t+1}, \bar{a}_{t+1})$
      Update $\theta_c^{\psi}$ using SGD($\alpha$) with $\mathcal{L}_{\psi} = (y - \tilde{\psi}_c(s_t, a_t))^2$
    $s_t \leftarrow s_{t+1}$

---

### C.2.4   SFRQL

The SFRQL agents (Algorithms 3, 4) allow to reevaluate policies in tasks with general reward functions. If the reward function is not given, an approximation $\tilde{R}_i$ of the reward function for each task $M_i$ is learned. The parameters for the approximation are randomly initialized at the beginning of each task. After each observed transition the approximation is updated according to the following loss:

$$\mathcal{L}_R(\theta_i^R) = \mathbb{E}\left\{ \left(r(s_t, a_t, s_{t+1}) - \tilde{R}_i(\phi(s_t, a_t, s_{t+1}); \theta_i^R)\right)^2 \right\} \tag{45}$$

In the case of tasks with linear reward functions the reward approximation becomes $\tilde{R}_i(\phi(s_t, a_t, s_{t+1}); \theta_i^R) = \phi(s_t, a_t, s_{t+1})^{\top} \theta_i^R$. Thus with $\theta_i^R = \tilde{\mathbf{w}}_i$ we recover the same procedure as for SF (43). For non-linear, general reward functions we represented $\tilde{R}$ with a neural network. The input of the network is the feature $\phi(s_t, a_t, s_{t+1})$. The network has one hidden layer with 10 neurons having ReLu activations. The output is a linear mapping to the scalar reward $r_t \in \mathbb{R}$.

All SFRQL agents learn an approximation of the $\tilde{\xi}_i$-function for each task $M_i$. Analogous to SF, the parameters of the $\tilde{\xi}$-function for the first task $\theta_1^{\xi}$ are randomly initialized. For consecutive tasks, they are initialized by copying them from the previous task ($\theta_i^{\xi} \leftarrow \theta_{i-1}^{\xi}$). The $\tilde{\xi}_i$-function of the current task $M_i$ is updated after each observed transition with the loss given

in (41). The SFRQL agents differ in their setting for $p(\phi_t = \phi|s_t, a_t)$ in the updates which is described in the upcoming sections. Besides the current $\tilde{\xi}_i$-function, the SFRQL agents also update the $\tilde{\xi}_c$-function which provided the GPI optimal action for the current transition: $c = \arg\max_{k \in \{1,2,...,i\}} \max_{a_t} \sum_{\phi \in \Phi} \tilde{\xi}_k(s_t, a_t, \phi) \tilde{R}_i(\phi)$. The update uses the same loss as for the update of the active $\tilde{\xi}_i$-function (41), but instead of using the GPI optimal action as next action, it uses the optimal action according to its own policy: $\bar{a}_{t+1} = \max_a \sum_{\phi \in \Phi} \tilde{\xi}_c(s_{t+1}, a, \phi) \tilde{R}_c(\phi)$.

**MF SFRQL (SFR):** The model-free SFRQL agent (Algorithm 3) uses a stochastic update for the $\tilde{\xi}$-functions. Given a transition, we set $p(\phi_t = \phi|s_t, a_t) \equiv 1$ for the observed feature $\phi = \phi(s_t, a_t, s_{t+1})$ and $p(\phi_t = \phi|s_t, a_t) \equiv 0$ for all other features $\phi \neq \phi(s_t, a_t, s_{t+1})$.

**MB SFRQL (MB SFR):** The one-step SF model-based SFRQL agent (Algorithm 4) uses an approximated model $\tilde{p}$ to predict $p(\phi_t = \phi|s_t, a_t)$ to reduce the variance of the $\xi$-function update. The model is by a linear mapping for each action. It uses a softmax activation to produce a valid distribution over $\Phi$:

$$\tilde{p}(s, a, \phi; \theta^p) = \frac{\exp(s^\top \theta^p_{a,\phi})}{\sum_{\phi' \in \Phi} \exp(s^\top \theta^p_{a,\phi'})} \tag{46}$$

where $\theta^p_{a,\phi} \in \mathbb{R}^{113}$. As $\tilde{p}$ is valid for each task in $\mathcal{M}$, its weights $\theta^p$ are only randomly initialized at the beginning of the first task. For each observed transition, the model is updated using the following loss:

$$\forall \phi \in \Phi : \ \mathcal{L}_p(\theta^p_i) = \mathbb{E}\left\{(p(\phi_t = \phi|s_t, a_t) - \tilde{p}(s_t, a_t, \phi; \theta^p_i))^2\right\} , \tag{47}$$

where we set $p(\phi_t = \phi|s_t, a_t) \equiv 1$ for the observed feature $\phi = \phi(s_t, a_t, s_{t+1})$ and $p(\phi_t = \phi|s_t, a_t) \equiv 0$ for all other features $\phi \neq \phi(s_t, a_t, s_{t+1})$.

**Continuous SFRQL (CSFR):** Under the condition of approximating a feature representation (CSFR-$h$), the feature space becomes continuous. In this case we are using a discretized version of the SFRQL algorihm that is explained in detail under Appendix D.2.4. The agent uses for the object collection environment a linear mapping to approximate the $\xi$-values given the observation $s$.

## C.3 Experimental Procedure

All agents were evaluated in both task types (linear or general reward function) on 300 tasks. The agents experienced the tasks sequentially, each for 20.000 steps. The agents had knowledge when a task change happened. Each agent was evaluated for 10 repetitions to measure their average performance. Each repetition used a different random seed that impacted the following elements: a) the sampling of the tasks, b) the random initialization of function approximator parameters, c) the stochastic behavior of the environments when taking steps, and d) the $\epsilon$-greedy action selection of the agents. The tasks, i.e. the reward functions, were different between the repetitions of a particular agent, but identical to the same repetition of a different agent. Thus, all algorithms were evaluated over the same tasks.

The SF agents (SF, SFRQL) were evaluated under two conditions regarding their reward model of a task. First, the reward weights or the reward function is given to them. Second, that they have to learn the reward weights or the reward function online during the training (SF-R, SF-$h$, SFR-R, CSFR-$h$). As the SF does not support general reward functions, it is not possible to provide the SF agent with the reward function in the first condition. As a solution, before the agent was trained on a new task $\mathcal{M}_i$, a linear model of the reward $R_i(\phi) = \phi^\top \tilde{w}_i$ was fitted. The initial approximation $\tilde{w}_i$ was randomly initialized and then fitted for 10.000 iterations using a gradient descent procedure based on the absolute mean error (L1 norm):

$$\Delta \tilde{w}_i = \eta \frac{1}{|\Phi|} \sum_{\phi \in \Phi} R_i(\phi) - \phi^\top \tilde{w}_i , \tag{48}$$

with a learning rate of $\eta = 1.0$ that yielded the best results tested over several learning rates.

---

**Algorithm 3:** Model-free SFRQL (SFR)

---

**Input:** exploration rate: $\epsilon$

learning rate for $\xi$-functions: $\alpha$

learning rate for reward models $R$: $\alpha_R$

features $\phi$ or $\tilde{\phi}$

optional: reward functions for tasks: $\{\tilde{R}_1, \tilde{R}_2, \ldots, \tilde{R}_{\text{num\_tasks}}\}$

---

**for** $i \leftarrow 1$ **to** *num_tasks* **do**

  **if** $\tilde{R}_i$ *not provided* **then** initialize $\tilde{R}_i$: $\theta_i^R \leftarrow$ small random initial values

  **if** $i = 1$ **then** initialize $\tilde{\xi}_i$: $\theta_i^\xi \leftarrow$ small random initial values **else** $\theta_i^\xi \leftarrow \theta_{i-1}^\xi$

  new_episode $\leftarrow$ true

  **for** $t \leftarrow 1$ **to** num_steps **do**

    **if** *new_episode* **then**

      new_episode $\leftarrow$ false

      $s_t \leftarrow$ initial state

    $c \leftarrow \arg\max_{k \in \{1,2,\ldots,i\}} \max_a \sum_\phi \tilde{\xi}_k(s_t, a, \phi) \tilde{R}_i(\phi)$   // GPI optimal policy

    With probability $\epsilon$ select a random action $a_t$, otherwise

     $a_t \leftarrow \arg\max_a \sum_\phi \tilde{\xi}_c(s_t, a, \phi) \tilde{R}_i(\phi)$

    Take action $a_t$ and observe reward $r_t$ and next state $s_{t+1}$

    **if** $\tilde{R}_i$ *not provided* **then**

      Update $\theta_i^R$ using SGD($\alpha_R$) with $\mathcal{L}_R = (r_t - \tilde{R}_i(\phi(s_t, a_t, s_{t+1})))^2$

    **if** $s_{t+1}$ *is a terminal state* **then**

      new_episode $\leftarrow$ true

      $\gamma_t \leftarrow 0$

    **else**

      $\gamma_t \leftarrow \gamma$

    // GPI optimal next action for task $i$

    $\bar{a}_{t+1} \leftarrow \arg\max_a \arg_{k \in \{1,2,\ldots,i\}} \sum_\phi \tilde{\xi}_k(s_{t+1}, a, \phi) \tilde{R}_i(\phi)$

    **foreach** $\phi \in \Phi$ **do**

      **if** $\phi = \phi(s_t, a_t, s_{t+1})$ **then** $y_\phi \leftarrow 1 + \gamma_t \tilde{\xi}_i(s_{t+1}, \bar{a}_{t+1}, \phi)$

      **else** $y_\phi \leftarrow \gamma_t \tilde{\xi}_i(s_{t+1}, \bar{a}_{t+1}, \phi)$

    Update $\theta_i^\xi$ using SGD($\alpha$) with $\mathcal{L}_\xi = \sum_\phi (y_\phi - \tilde{\xi}_i(s_t, a_t, \phi))^2$

    **if** $c \neq i$ **then**

      // optimal next action for task $c$

      $\bar{a}_{t+1} \leftarrow \arg\max_a \sum_\phi \tilde{\xi}_c(s_{t+1}, a, \phi) \tilde{R}_c(\phi)$

      **foreach** $\phi \in \Phi$ **do**

        **if** $\phi = \phi(s_t, a_t, s_{t+1})$ **then** $y_\phi \leftarrow 1 + \gamma_t \tilde{\xi}_c(s_{t+1}, \bar{a}_{t+1}, \phi)$

        **else** $y_\phi \leftarrow \gamma_t \tilde{\xi}_c(s_{t+1}, \bar{a}_{t+1}, \phi)$

      Update $\theta_c^\xi$ using SGD($\alpha$) with $\mathcal{L}_\xi = \sum_\phi (y_\phi - \tilde{\xi}_c(s_t, a_t, \phi))^2$

    $s_t \leftarrow s_{t+1}$

---

---

**Algorithm 4:** One Step SF-Model SFRQL (MB SFR)

---

**Input:** exploration rate: $\epsilon$
      learning rate for $\xi$-functions: $\alpha$
      learning rate for reward models $R$: $\alpha_R$
      learning rate for the one-step SF model $\tilde{p}$: $\beta$
      features $\phi$ or $\tilde{\phi}$
      optional: reward functions for tasks: $\{\tilde{R}_1, \tilde{R}_2, \ldots, \tilde{R}_{\text{num\_tasks}}\}$

initialize $\tilde{p}$: $\theta^p \leftarrow$ small random initial values
**for** $i \leftarrow 1$ **to** *num_tasks* **do**
    **if** $\tilde{R}_i$ *not provided* **then** initialize $\tilde{R}_i$: $\theta_i^R \leftarrow$ small random initial values
    **if** $i = 1$ **then** initialize $\tilde{\xi}_i$: $\theta_i^\xi \leftarrow$ small random initial values **else** $\theta_i^\xi \leftarrow \theta_{i-1}^\xi$
    new_episode $\leftarrow$ true
    **for** $t \leftarrow 1$ **to** num_steps **do**
        **if** *new_episode* **then**
            new_episode $\leftarrow$ false
            $s_t \leftarrow$ initial state
        $c \leftarrow \arg\max_{k \in \{1,2,\ldots,i\}} \max_a \sum_\phi \tilde{\xi}_k(s_t, a, \phi)\tilde{R}_i(\phi)$    `// GPI optimal policy`
        With probability $\epsilon$ select a random action $a_t$, otherwise
          $a_t \leftarrow \arg\max_a \sum_\phi \tilde{\xi}_c(s_t, a, \phi)\tilde{R}_i(\phi)$
        Take action $a_t$ and observe reward $r_t$ and next state $s_{t+1}$
        **if** $\tilde{R}_i$ *not provided* **then**
            Update $\theta_i^R$ using SGD($\alpha_R$) with $\mathcal{L}_R = (r_t - \tilde{R}_i(\phi(s_t, a_t, s_{t+1})))^2$
        **foreach** $\phi \in \Phi$ **do**
            **if** $\phi = \phi(s_t, a_t, s_{t+1})$ **then** $y_\phi \leftarrow 1$ **else** $y_\phi \leftarrow 0$
        Update $\theta^p$ using SGD($\beta$) with $\mathcal{L}_p = \sum_\phi (y_\phi - \tilde{p}(s_t, a_t, \phi))^2$
        **if** $s_{t+1}$ *is a terminal state* **then**
            new_episode $\leftarrow$ true
            $\gamma_t \leftarrow 0$
        **else**
            $\gamma_t \leftarrow \gamma$
        `// GPI optimal next action for task i`
        $\bar{a}_{t+1} \leftarrow \arg\max_a \arg_{k \in \{1,2,\ldots,i\}} \sum_\phi \tilde{\xi}_k(s_{t+1}, a, \phi)\tilde{R}_i(\phi)$
        **foreach** $\phi \in \Phi$ **do**
            $y_\phi \leftarrow \tilde{p}(s_t, a_t, \phi) + \gamma_t \tilde{\xi}_i(s_{t+1}, \bar{a}_{t+1}, \phi)$
        Update $\theta_i^\xi$ using SGD($\alpha$) with $\mathcal{L}_\xi = \sum_\phi (y_\phi - \tilde{\xi}_i(s_t, a_t, \phi))^2$
        **if** $c \neq i$ **then**
            `// optimal next action for task c`
            $\bar{a}_{t+1} \leftarrow \arg\max_a \sum_\phi \tilde{\xi}_c(s_{t+1}, a, \phi)\tilde{R}_c(\phi)$
            **foreach** $\phi \in \Phi$ **do**
               $y_\phi \leftarrow \tilde{p}(s_t, a_t, \phi) + \gamma_t \tilde{\xi}_c(s_{t+1}, \bar{a}_{t+1}, \phi)$
            Update $\theta_c^\xi$ using SGD($\alpha$) with $\mathcal{L}_\xi = \sum_\phi (y_\phi - \tilde{\xi}_c(s_t, a_t, \phi))^2$
        $s_t \leftarrow s_{t+1}$

**Hyperparameters:** The hyperparameters of the algorithms were set to the same values as in (Barreto et al., 2017). A grid search over the learning rates of all algorithms was performed. Each learning rate was evaluated for three different settings which are listed in Table 1. If algorithms had several learning rates, then all possible combinations were evaluated. This resulted in a different number of evaluations per algorithm and condition: Q- 3, SF- 3, SF-R - 9, SF-4 - 9, SF-8 - 9, SFR-3, SFR-R - 9, CSFR-3 - 9, MB SFR- 9, MB SFR-R - 27. In total, 100 parameter combinations were evaluated. The reported performances in the figures are for the parameter combination that resulted in the highest cumulative total reward averaged over all 10 repetitions in the respective environment. Please note, the learning rates $\alpha$ and $\alpha_{\mathbf{w}}$ are set to half of the rates defined in (Barreto et al., 2017). This is necessary due to the differences in calculating the loss and the gradients in the current paper. We use mean squared error loss formulations, whereas (Barreto et al., 2017) uses absolute error losses. The probability for random actions of the $\epsilon$-Greedy action selection was set to $\epsilon = 0.15$ and the discount rate to $\gamma = 0.95$. The initial weights $\theta$ for the function approximators were randomly sampled from a standard distribution with $\theta_{\text{init}} \sim \mathcal{N}(\mu = 0, \sigma = 0.01)$.

Table 1: Evaluated Learning Rates in the Object Collection Environment

| Parameter | Description | Values |
|---|---|---|
| $\alpha$ | Learning rate of the Q, $\psi$, and $\xi$-function | $\{0.0025, 0.005, 0.025\}$ |
| $\alpha_{\mathbf{w}}, \alpha_R$ | Learning rate of the reward weights or the reward model | $\{0.025, 0.05, 0.075\}$ |
| $\beta$ | Learning rate of the One-Step SF Model | $\{0.2, 0.4, 0.6\}$ |

**Computational Resources and Performance:** Experiments were conducted on a cluster with a variety of node types (Xeon SKL Gold 6130 with 2.10GHz, Xeon SKL Gold 5218 with 2.30GHz, Xeon SKL Gold 6126 with 2.60GHz, Xeon SKL Gold 6244 with 3.60GHz, each with 192 GB Ram, no GPU). The time for evaluating one repetition of a certain parameter combination over the 300 tasks depended on the algorithm and the task type. Linear reward function tasks: $Q \approx 1h$, $SF \approx 4h$, $SF\text{-}R \approx 4h$, $SF\text{-}4 \approx 4h$, $SF\text{-}8 \approx 4h$, $SFR \approx 15h$, $SFR\text{-}R \approx 42h$, $CSFR\text{-}4 \approx 14h$, $MB\ SFR \approx 16h$, and $MB\ SFR\text{-}R \approx 43h$. General reward function tasks: $Q \approx 1h$, $SF \approx 5h$, $SF\text{-}R \approx 4h$, $SF\text{-}4 \approx 4h$, $SF\text{-}8 \approx 5h$, $SFR \approx 14h$, $SFR\text{-}R \approx 68h$, $CSFR\text{-}4 \approx 13h$, $MB\ SFR \approx 18h$, and $MB\ SFR\text{-}R \approx 67h$. Please note, the reported times do not represent well the computational complexity of the algorithms, as the algorithms were not optimized for speed, and some use different software packages (numpy or pytorch) for their individual computations.

## C.4 EFFECT OF INCREASING NON-LINEARITY IN GENERAL REWARD TASK

We further studied the effect of general reward functions on the performance of classical SF compared to SFRQL (Fig. 1, c). We evaluated the agents in tasks with different levels of difficulty in relation to how well their reward functions can be approximated by a linear model. Seven difficulty levels have been evaluated. For each level, the agents were trained sequentially on 300 tasks as for the experiments with general reward functions. The reward functions for each level were sampled with the following procedure. Several general reward functions were randomly sampled as previously described. For each reward function, a linear model of a reward weight vector $\tilde{\mathbf{w}}$ was fitted using the same gradient descent procedure as in Eq. 48. The final average absolute model error after 10.000 iterations was measured. Each of the seven difficulty levels defines a range of model errors its tasks have with the following increasing ranges: $\{[0.0, 0.25], [0.25, 0.5], \ldots, [1.5, 1.75]\}$. For each difficulty level, 300 reward functions were selected that yield a linear model are in the respective range of the level.

Q, SF, and SFR were each trained on 300 tasks, i.e. reward functions, on each difficulty level. As hyperparameters were the best performing parameters from the previous general reward task experiments used. We measured the ratio between the total return over 300 tasks of Q-learning and MF SFRQL (Q/SFR), and SF and MF SFRQL (SF/SFR). Fig. 1 (c) shows the results, using as x-axis the mean average absolute model error defined by the bracket of each difficulty level. The results show that the relative performance of SF compared to SFR reduces with higher non-linearity of the reward functions. For reward functions that are nearly linear (mean error of 0.125), both have a similar performance. Whereas, for reward functions that are difficult to model with a linear relation (mean error of 1.625) SF reaches only less than $50\%$ of the performance of MF SFRQL.

## D  EXPERIMENTAL DETAILS: RACER ENVIRONMENT

This section extends the brief introduction to the racer environment (Fig. 2, a) given in Section 4.2.

### D.1  ENVIRONMENT

The environment is a continuous two-dimensional area in which the agent drives similar to a car. The position of the agent is a point in the 2D space: $p = (x, y) \in [0, 1]^2$. Moreover, the agent has an orientation which it faces: $\theta \in [-\pi, \pi 1]$. The action space of the agent consists of three movement directions: $A = \{$right, straight, left $\}$. Each action changes the position of the agent depending on its current position and orientation. The action *straight* changes the agent's position by $0.075$ towards its orientation $\theta$. The action *right* changes the orientation of the agent to $\theta + \frac{1}{7}\pi$ and its position $0.06$ towards this new direction, whereas *left* the direction to $\theta - \frac{1}{7}\pi$ changes. The environment is stochastic by adding Gaussian noise with $\sigma = 0.005$ to the final position $x$, $y$, and orientation $\theta$. If the agent drives outside the area $(x, y) \in [0, 1]^2$, then it reappears on the other opposite side. The environment resembles therefore a torus (or donut). As a consequence, distances $d(p_x, p_y)$ are also measure in this space, so that the positions $p_x = (0.1, 0.5)$ and $p_y = (0.9, 0.5)$ have not a distance of $0.8$ but $d(p_x, p_y) = 0.2$. The environment has 3 markers at the positions $m_1 = (0.25, 0.75)$, $m_2 = (0.75, 0.25)$, and $m_3 = (0.75, 0.6)$. The features measure the distance of the agent to each marker: $\phi \in \mathbb{R}^3$ with $\phi_k = d(p, m_k)$. Each feature dimensions is normalized to be $\phi_k \in [0, 1]$. At the beginning of an episode the agent is randomly placed and oriented in environment. An episode ends after $200$ time steps.

The state space of the agents is similarly constructed as for the object collection environment. The agent's position is encoded with a $10 \times 10$ radial basis functions $s_{\text{pos}} \in \mathbb{R}^{100}$ as defined in 39. In difference, that the distances are measure according to the torus shape. A similar radial basis function approach is also used to encode the orientation $s_{\text{ori}} \in \mathbb{R}^{20}$ of the agent using 20 equally distributed gaussian centers in $[-\pi, \pi]$ and $\sigma = \frac{1}{5}\pi$. Please note, $\pi$ and $-\pi$ are also connected in this space, i.e. $d(\pi, -\pi) = 0$. The combination of the position and orientation of the agent is the final state: $s = [s_{\text{pos}}^\top, s_{\text{ori}}^\top]^\top \in \mathbb{R}^{120}$.

The reward functions define preferred positions in the environment based on the features, i.e. the distance of the agent to the markers. A preference function $r_k$ exists for each distance. The functions are composed of a maximization over $m$ Gaussian components that evaluate the agents distance:

$$R(\phi) = \sum_{k=1}^{3} r_k(\phi_k) \quad \text{with} \quad r_i = \frac{1}{3} \max \left\{ \exp\left( -\frac{(\phi_k - \mu_j)^2}{\sigma_j} \right) \right\}_{j=1}^{m} . \tag{49}$$

Reward functions are randomly generated by sampling the number of Gaussian components $m$ to be 1 or 2. The properties of each component are sampled according to $\mu_j \sim \mathcal{U}(0.0, 0.7)$, and $\sigma_j \sim \mathcal{U}(0.001, 0.01)$. Fig. 2 (a) illustrates one such randomly sampled reward function where dark areas represent locations with high rewards.

### D.2  ALGORITHMS

We evaluated Q, SF, SF-R, SF-$h$ (see Appendix C.2 for their full description) and CSFR, CSFR-R, and CSFR-$h$ in the racer environment. In difference to their implementation for the object collection environment, they used a different neural network architecture to approximate their respective value functions.

#### D.2.1  LEARNING OF FEATURE REPRESENTATIONS

Similar to the object collection environment, learned features were also evaluated for the racer environment with the procedure described in Section C.2.1. In difference, the dataset $\mathcal{D}'$ consisted of transitions from the Q agent from the initial 10 tasks of the racer environment. Moreover, zero reward transitions were not filtered as the rewards are more dense.

### D.2.2   QL

Q-learning uses a fully connected feedforward network with bias and a ReLU activation for hidden layers. It has 2 hidden layers with 20 neurons each.

### D.2.3   SFQL

SFQL uses a feedforward network with bias and a ReLU activation for hidden layers. It has for each of the three feature dimensions a separate fully connected subnetwork. Each subnetwork has 2 hidden layers with 20 neurons each.

### D.2.4   CONTINUOUS MODEL-FREE SFRQL

The racer environment has continuous features $\phi \in \mathbb{R}^3$. Therefore, the MF SFRQL procedure (Alg. 3) can not be directly applied as it is designed for discrete feature spaces. We introduce here a MF SFRQL procedure for continuous feature spaces (CSFR). It is a feature dimension independent, and discretized version of SFRQL. As the reward functions (49) are a sum over the individual feature dimensions, the Q-value can be computed as:

$$Q^{\pi}(s,a) = \int_{\Phi} R(\phi)\xi^{\pi}(s,a,\phi)\mathrm{d}\phi = \sum_{k}\int_{\Phi_k} r_k(\phi_k)\xi_k^{\pi}(s,a,\phi_k)\mathrm{d}\phi_k \ , \tag{50}$$

where $\Phi_k$ is the feature space for each feature dimension which is $\Phi_k = [0,1]$ in the racer environment. $\xi_k^{\pi}$ is a $\xi$-function for the feature dimension $k$. (50) shows that the $\xi$-function can be independently represented over each individual feature dimension $\phi_k$, instead of over the full features $\phi$. This reduces the complexity of the approximation.

Moreover, we introduce a discretization of the $\xi$-function that discretizes the space of each feature dimension $k$ in $U = 11$ bins with the centers:

$$X_k = \left\{\phi_k^{\min} + j\Delta\phi_k : 0 < j < U\right\}, \ \ \text{with} \ \ \Delta\phi_k := \frac{\phi_k^{max} - \phi_k^{min}}{U-1} \ ,$$

where $\Delta\phi_k$ is the distance between the centers, and $\phi_k^{min} = 0.0$ is the lowest center, and $\phi_k^{max} = 1.0$ the largest center. Given this discretization and the decomposition of the Q-function according to (50), the Q-values can be computed by:

$$Q^{\pi}(s,a) = \sum_{k}\sum_{x \in X_k} R(x)\xi^{\pi}(s,a,x) \ .$$

Alg. 5 lists the complete CSFR procedure with the update steps for the $\xi$-functions. Similar to the SF agent, CSFR uses a feedforward network with bias and a ReLU activation for hidden layers. It has for each of the three feature dimensions a separate fully connected subnetwork. Each subnetwork has 2 hidden layers with 20 neurons each. The discretized outputs per feature dimension share the last hidden layer per subnetwork.

The $\xi$-function is updated according to the following procedure. Instead of providing a discrete learning signal to the model, we encode the observed feature using continuous activation functions around each bin center. Given the $j$'th bin center of dimension $k$, $x_{k,j}$, its value is encoded to be $1.0$ if the feature value of this dimension aligns with the center ($\phi_k = x_{k,j}$). Otherwise, the encoding for the bin decreases linearly based on the distance between the bin center and the value ($|x_{k,j} - \phi_k|$) and reaches 0 if the value is equal to a neighboring bin center, i.e. has a distance $\geq \Delta\phi_k$. We represent this encoding for each feature dimension $k$ by $\mathbf{u}_k \in (0,1)^U$ with:

$$\forall_{k \in \{1,2,3\}} : \forall_{0<j<U} : u_{k,j} = \max\left(0, \frac{(1 - |x_{k,j} - \phi_k|)}{\Delta\phi_k}\right) \ .$$

To learn $\tilde{\xi}$ we update the parameters $\theta^{\xi}$ using stochastic gradient descent following the gradients $\nabla_{\theta^{\xi}}\mathcal{L}_{\xi}(\theta^{\xi})$ of the loss based on the SFRQL update (13):

$$\forall\phi \in \Phi : \ \mathcal{L}_{\xi}(\theta^{\xi}) = \mathbb{E}\left\{\frac{1}{n}\sum_{k=1}^{3}\left(\mathbf{u}_k + \gamma\tilde{\xi}_k(s_{t+1},\bar{a}_{t+1};\bar{\theta}^{\xi}) - \tilde{\xi}_k(s_t,a_t;\theta^{\xi})\right)^2\right\}$$

$$\text{with} \ \bar{a}_{t+1} = \arg\max_{a}\sum_{k}\sum_{x \in X_k} R(x)\tilde{\xi}(s_{t+1},a,\phi;\bar{\theta}^{\xi}) \ , \tag{51}$$

---

**Algorithm 5:** Model-free SFRQL for Continuous Features (CSFR)

---

**Input:** exploration rate: $\epsilon$

learning rate for $\xi$-functions: $\alpha$

learning rate for reward models $R$: $\alpha_R$

features $\phi$ or $\tilde{\phi} \in \mathbb{R}^n$

components of reward functions for tasks: $\{R_1 = \{r_1^1, r_2^1, ..., r_n^1\}, R_2, \ldots, R_{\text{num\_tasks}}\}$

discretization parameters: $X, \Delta\phi$

---

**for** $i \leftarrow 1$ **to** *num_tasks* **do**

  **if** $i = 1$ **then**

    $\forall_{k \in \{1,...,n\}}$: initialize $\tilde{\xi}_k^i$: $\theta_{i,k}^\xi \leftarrow$ small random values

  **else**

    $\forall_{k \in \{1,...,n\}}$: $\theta_{i,k}^\xi \leftarrow \theta_{i-1,k}^\xi$

  new_episode $\leftarrow$ true

  **for** $t \leftarrow 1$ **to** num_steps **do**

    **if** *new_episode* **then**

      new_episode $\leftarrow$ false

      $s_t \leftarrow$ initial state

    $c \leftarrow \arg\max_{j \in \{1,2,...,i\}} \max_a \sum_{k=1}^n \sum_{x \in X_k} \tilde{\xi}_k^j(s_t, a, x) r_k^i(x)$    // GPI policy

    With probability $\epsilon$ select a random action $a_t$, otherwise

     $a_t \leftarrow \arg\max_a \sum_{k=1}^n \sum_{x \in X_k} \tilde{\xi}_k^j(s_t, a, x) r_k^i(x)$

    Take action $a_t$ and observe reward $r_t$ and next state $s_{t+1}$

    **if** $s_{t+1}$ *is a terminal state* **then**

      new_episode $\leftarrow$ true

      $\gamma_t \leftarrow 0$

    **else**

      $\gamma_t \leftarrow \gamma$

    // GPI optimal next action for task $i$

    $\bar{a}_{t+1} \leftarrow \arg\max_a \arg_{j \in \{1,2,...,i\}} \sum_{k=1}^n \sum_{x \in X_k} \tilde{\xi}_k^j(s_t, a, x) r_k^i(x)$

    $\phi_t \leftarrow \phi(s_t, a_t, s_{t+1})$

    **for** $k \leftarrow 1$ **to** $n$ **do**

      **foreach** $x \in X_k$ **do**

        $y_{k,x} \leftarrow \max\left(0, 1 - \frac{|x - \phi_{t,k}|}{\Delta\phi}\right) + \gamma_t \tilde{\xi}_k^i(s_{t+1}, \bar{a}_{t+1}, x)$

    Update $\theta_i^\xi$ using SGD($\alpha$) with $\mathcal{L}_\xi = \sum_{k=1}^n \sum_{x \in X_k} (y_{k,x} - \tilde{\xi}_k^i(s_t, a_t, x))^2$

    **if** $c \neq i$ **then**

      // optimal next action for task $c$

      $\bar{a}_{t+1} \leftarrow \arg\max_a \sum_{k=1}^n \sum_{x \in X_k} \tilde{\xi}_k^c(s_t, a, x) r_k^c(x)$

      **for** $k \leftarrow 1$ **to** $n$ **do**

        **foreach** $x \in X_k$ **do**

          $y_{k,x} \leftarrow \max\left(0, 1 - \frac{|x - \phi_{t,k}|}{\Delta\phi}\right) + \gamma_t \tilde{\xi}_k^c(s_{t+1}, \bar{a}_{t+1}, x)$

      Update $\theta_c^\xi$ using SGD($\alpha$) with $\mathcal{L}_\xi = \sum_{k=1}^n \sum_{x \in X_k} (y_{k,x} - \tilde{\xi}_k^c(s_t, a_t, x))^2$

    $s_t \leftarrow s_{t+1}$

where $n = 3$ is the number of feature dimensions and $\tilde{\xi}_k$ is the vector of the $U$ discretized $\xi$-values for dimension $k$.

## D.3 EXPERIMENTAL PROCEDURE

All agents were evaluated on 40 tasks. The agents experienced the tasks sequentially, each for 1000 episodes ($200, 000$ steps per task). The agents had knowledge when a task change happened. Each agent was evaluated for 10 repetitions to measure their average performance. Each repetition used a different random seed that impacted the following elements: a) the sampling of the tasks, b) the random initialization of function approximator parameters, c) the stochastic behavior of the environments when taking steps, and d) the $\epsilon$-greedy action selection of the agents. The tasks, i.e. the reward functions, were different between the repetitions of a particular agent, but identical to the same repetition of a different agent. Thus, all algorithms were evaluated over the same tasks.

SF was evaluated under two conditions. First, by learning reward weights $\tilde{\mathbf{w}}_i$ with the iterative gradient decent method in (48). The weights were trained for $10, 000$ iterations with an learning rate of 1.0. At each iteration, 50 random points in the task were sampled and their features and rewards are used for the training step. Second, by learning the reward weights online during the training (SF-R).

**Hyperparameters** A grid search over the learning rates of all algorithms was performed. Each learning rate was evaluated for three different settings which are listed in Table 2. If algorithms had several learning rates, then all possible combinations were evaluated. This resulted in a different number of evaluations per algorithm and condition: Q- 4, SF- 4, SF-R - 12, SF-3 - 12, SF-6 - 12, CSFR- 4, CSFR-R - 12, CSFR-3 - 12. In total, 72 parameter combinations were evaluated. The reported performances in the figures are for the parameter combination that resulted in the highest cumulative total reward averaged over all 10 repetitions in the respective environment. The probability for random actions of the $\epsilon$-Greedy action selection was set to $\epsilon = 0.15$ and the discount rate to $\gamma = 0.9$. The initial weights and biases $\theta$ for the function approximators were initialized according to an uniform distribution with $\theta_i \sim \mathcal{U}(-\sqrt{k}, \sqrt{k})$, where $k = \frac{1}{\text{in\_features}}$.

Table 2: Evaluated Learning Rates in the Racer Environment

| Parameter | Description | Values |
|:---:|:---|:---:|
| $\alpha$ | Learning rate of the Q, $\psi$, and $\xi$-function | $\{0.0025, 0.005, 0.025, 0.5\}$ |
| $\alpha_{\mathbf{w}}$ | Learning rate of the reward weights | $\{0.025, 0.05, 0.075\}$ |

**Computational Resources and Performance:** Experiments were conducted on the same cluster as for the object collection environment experiments. The time for evaluating one repetition of a certain parameter combination over the 40 tasks depended on the algorithm: Q $\approx 9h$, SF $\approx 73h$, SF-R $\approx 70h$, SF-3 $\approx 20h$, SF-6 $\approx 19h$, CSFR $\approx 88h$, CSFR-R $\approx 80h$, and CSFR-3 $\approx 75h$. Please note, the reported times do not represent well the computational complexity of the algorithms, as the algorithms were not optimized for speed, and some use different software packages (numpy or pytorch) for their individual computations.

# E  ADDITIONAL EXPERIMENTAL RESULTS

This section reports additional results and experiments:

1. Evaluation of the agents in the object collection task by Barreto et al. (2017)
2. Report of the total return and the statistical significance of differences between agents for all experiments

## E.1  OBJECT COLLECTION TASK BY BARRETO ET AL. (2017)

We additionally evaluated all agents in the object collection task by Barreto et al. (2017).

**Environment:**  The environment differs to the modified object collection task (Appendix C) only in terms of the objects and features. The environment has 3 object types: orange, blue, and pink (Fig. 3). The feature encode if the agent has collected one of these object types or if it reached the goal area. The first three dimensions of the features $\phi(s_t, a_t, s_{t+1}) \in \Phi \subset \{0,1\}^4$ encode which object type is collected. The last dimension encodes if the goal area was reached. In total $|\Phi| = 5$ possible features exists: $\phi_1 = [0,0,0,0]^\top$ - standard observation, $\phi_2 = [1,0,0,0]^\top$ - collected an orange object, $\phi_3 = [0,1,0,0]^\top$ - collected a blue object, $\phi_4 = [0,0,1,0]^\top$ - collected a pink object, and $\phi_5 = [0,0,0,1]^\top$ - reached the goal area. Agents were also evaluated with learned features that have either a dimension of $h = 4$ or $h = 8$. The features were learned according to the procedure described in Appendix C.2.1.

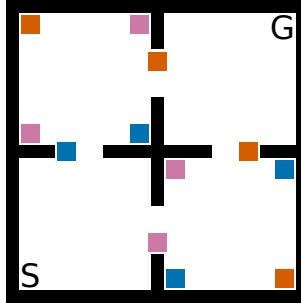

Figure 3:  Object collection environment from (Barreto et al., 2017) with 3 object types: orange, blue, pink.

The rewards $r = \phi^\top \mathbf{w}_i$ are defined by a linear combination of discrete features $\phi \in \mathbb{N}^4$ and a weight vector $\mathbf{w} \in \mathbb{R}^4$. The first three dimensions in $\mathbf{w}$ define the reward that the agent receives for collecting one of the object types. The final weight defines the reward for reaching the goal state which is $\mathbf{w}_4 = 1$ for each task. All agents were trained in on 300 randomly generated linear reward functions with the same experimental procedure as described in Appendix. C. For each task the reward weights for the 3 objects are randomly sampled from a uniform distribution: $\mathbf{w}_{k \in \{1,2,3\}} \sim \mathcal{U}(-1, 1)$.

**Results:**  The results (Fig. 4) follow closely the results from the modified object collection task (Fig. 1, b, and 5, a). MF SFRQL (SFR) reaches the highest performance outperforming SF in terms of learning speed and asymptotic performance. It is followed by MB SFR and SF which show no statistical significant difference between each other in their final performance. Nonetheless, MB $\xi$ has a higher learning speed during the initial 40 tasks. The results for the agents that learn the reward weights online (SF-R, SFR-R, and MB SFR-R) follow the same trend with SFR-R outperforming SF-R slightly. Nonetheless, the $\xi$-agents have a much stronger learning speed during the initial 70 tasks compared to SF-R, due to the errors in the approximation of the weight vectors, especially at the beginning of a new task. The agent with approximated features (SF-$h$, SFR-$h$) have a similar performance that is below the performance of all other SF agents. All SF agents can clearly outperform standard Q-learning.

Please note, the SF agents with with approximated features show a slighlty better performance in the experimental results by (Barreto et al., 2017) than in our experiments. (Barreto et al., 2017) does not provide the hyperparameters for the approximation procedure of the features. Therefore, a difference between their and our hyperparameters could explain the different results.

## E.2  TOTAL RETURN IN TRANSFER LEARNING EXPERIMENTS AND STATISTICAL SIGNIFICANT DIFFERENCES

Fig. 5 shows for each of the transfer learning experiments in the object collection and the racer environment the total return that each agent accumulated over all tasks. Each dot besides the boxplot shows the total return for each of the 10 repetitions. The box ranges from the upper to the lower

quartile. The whiskers represent the upper and lower fence. The mean and standard deviation are indicated by the dashed line and the median by the solid line. The tables in Fig. 5 report the p-value of pairwise Mann–Whitney U test. A significant different total return can be expected if $p < 0.05$.

For the object collection environment (Fig.5, a, b), SFRQL (SFR) outperforms SF in both conditions, in tasks with linear and general reward functions. However, the effect is stronger in tasks with general reward functions where SF has more problems to correctly approximate the reward function with its linear approach. For the conditions, where the agents learn a reward model online (SF-R, SFR-R, MB SFR-R) and where the features are approximated (SF-$h$, CSFR-$h$) the difference between the algorithms in the general reward case is not as strong due the effect of their poor approximated reward models.

In the racer environment (Fig.5, c) SF has a poor performance below standard Q-learning as it can not appropriately approximate the reward functions with a linear model. In difference CSFR outperforms Q. Also in this environment for the conditions, where the agents learn a reward model online (SF-R, CSFR-R) and where the features are approximated (SF-$h$, CSFR-$h$) the difference between the algorithms is not as strong due the effect of their poor approximated reward models.

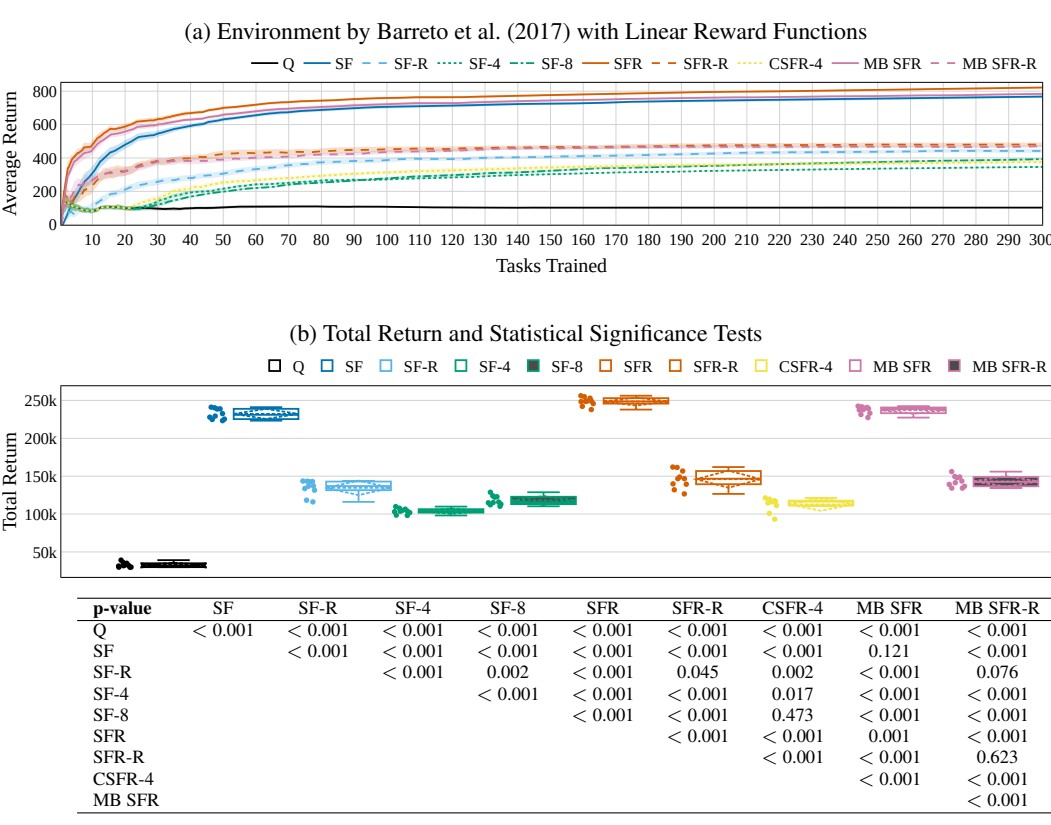

(a) Environment by Barreto et al. (2017) with Linear Reward Functions

(b) Total Return and Statistical Significance Tests

| p-value | SF | SF-R | SF-4 | SF-8 | SFR | SFR-R | CSFR-4 | MB SFR | MB SFR-R |
|---|---|---|---|---|---|---|---|---|---|
| Q | < 0.001 | < 0.001 | < 0.001 | < 0.001 | < 0.001 | < 0.001 | < 0.001 | < 0.001 | < 0.001 |
| SF | | < 0.001 | < 0.001 | < 0.001 | < 0.001 | < 0.001 | < 0.001 | 0.121 | < 0.001 |
| SF-R | | | < 0.001 | 0.002 | < 0.001 | 0.045 | 0.002 | < 0.001 | 0.076 |
| SF-4 | | | | < 0.001 | < 0.001 | < 0.001 | 0.017 | < 0.001 | < 0.001 |
| SF-8 | | | | | < 0.001 | < 0.001 | 0.473 | < 0.001 | < 0.001 |
| SFR | | | | | | < 0.001 | < 0.001 | 0.001 | < 0.001 |
| SFR-R | | | | | | | < 0.001 | < 0.001 | 0.623 |
| CSFR-4 | | | | | | | | < 0.001 | < 0.001 |
| MB SFR | | | | | | | | | < 0.001 |

Figure 4: MF SFRQL (SFR) outperforms SF in the object collection environment by Barreto et al. (2017), both in terms of asymptotic performance and learning speed. (a) The average over 10 runs of the average reward per task per algorithm and the standard error of the mean are depicted. (b) Total return over the 300 tasks in each evaluated condition. The table shows the p-values of pairwise Mann–Whitney U tests between the agents.

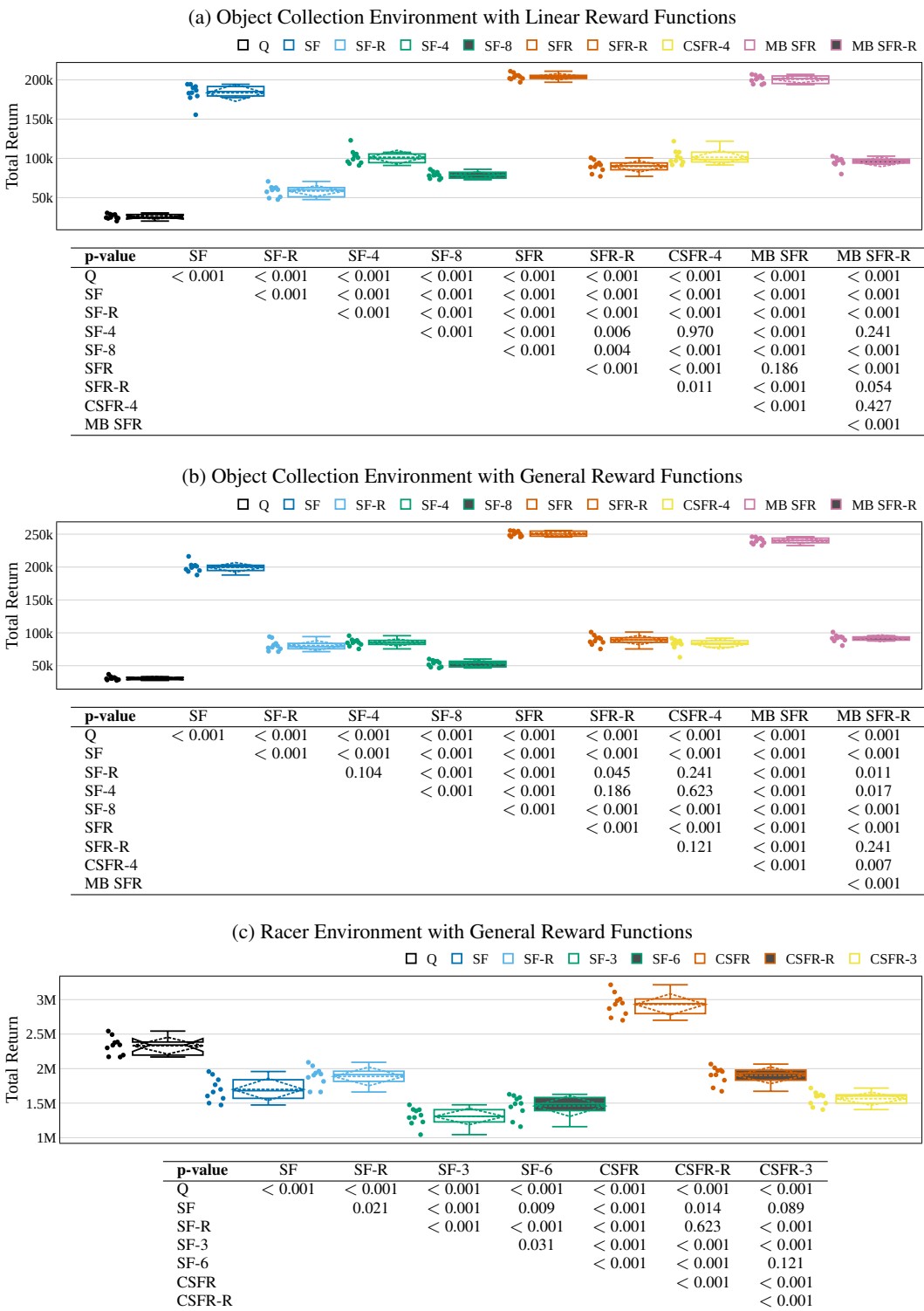

Figure 5: Total return over all tasks in each evaluated condition. The tables show the p-values of pairwise Mann–Whitney U tests between the agents. See the text for more information.

