# OpenReview forum: "Successor Feature Representations"
_ICLR.cc/2023/Workshop/RRL — RRL 2023 Poster_

### Official Review · Reviewer_WEBY · 2023-02-27
**Nice work to expand successor feature to more general case.**

**Rating:** 3
**Confidence:** 3

**Review:**

The authors have extended Barreto's work (SF) to a more general case that does not require the assumption that reward functions are not linearly decomposable. They propose the use of a xi function, which provides a guarantee of convergence. The xi function can be applied to a range of tasks that differ only in terms of their reward functions.

Experimental results demonstrate that the proposed model achieves higher returns than other methods with smaller number of tasks training.

Although the paper briefly mentions the complexity reduction achieved through the use of the xi function, further explanation and clarification on this point would be benefical.

---

### Official Review · Reviewer_sE8q · 2023-03-01
**Learning Value-Functions through non-linear transformations of Successor Features**

**Rating:** 2
**Confidence:** 5

**Review:**

The authors propose an extension to Successor Features that doesn’t require assuming the value function is a linear combination of the successor features $\phi$. To this end, they devise a TD-based method to learn a "Successor Feature Representation" which in tandem with a potentially non-linear reward function can be used to obtain a value function.

While the paper is well-written and the goal of addressing the shortcoming of Successor Features is laudable, there are a few shortcomings with the current version of the paper:


1. I believe there are some issues with certain derivations, in the case of discrete features this wouldn’t make a difference but it does affect the continuous case.
	- A lot of the derivations seem to be a little too loose with whether we’re dealing with an expectation or not. For example, the TD learning update (and proof) assume that we’re using a stochastic approximation algorithm to find the roots of $\mathbb{E} \left[ \sum_{t>0} \gamma^t \mathbf{1} \lbrace \phi_{t+1} = \phi \rbrace \right] - \xi(s, a, \phi) = 0$. When pushing $p(\phi | s, a)$ into the expectation the corresponding indicator oftentimes doesn't show up in the paper. For example, I believe the TD update should look like: $\xi(s_t, a_t, \phi) = \left(1 - \alpha \right) \xi(s_t, a_t, \phi) + \alpha \left( \mathbf{1} \lbrace \phi_{t+1} = \phi \rbrace + \gamma \xi(s_{t+1}, a_{t+1}, \phi) \right)$. As mentioned this wouldn't change the results in the paper for discrete features as the pdf is equivalent to the indicator.
	- I didn’t carefully look over the derivations so I’m unsure whether this omission impacts other aspects of this work.
2. The authors assume sufficient features are given, this usually isn’t true and SFs must be combined with some other method to learn a representation.
3. In the continuous case the proposed method is as intangible as learning the SR (this is alluded to as the proposed solution is to just discretize the continuous features), although this may be able to be overcome with recent work on the successor measure.

Despite these limitations, I believe with some revisions, this work has the potential to make a valuable contribution. The topic is relevant to the workshop and therefore I still lean toward acceptance.